# Gene-free methodology for cell fate dynamics during development

**Francis Corson[1]\*, Eric D Siggia[2]\***

[1]Laboratoire de Physique Statistique, CNRS / Ecole Normale Supérieure, Paris, France; [2]Center for Studies in Physics and Biology, Rockefeller University, New York, United States

**Abstract** Models of cell function that assign a variable to each gene frequently lead to systems of equations with many parameters whose behavior is obscure. Geometric models reduce dynamics to intuitive pictorial elements that provide compact representations for sparse in vivo data and transparent descriptions of developmental transitions. To illustrate, a geometric model fit to vulval development in *Caenorhabditis elegans*, implies a phase diagram where cell-fate choices are displayed in a plane defined by EGF and Notch signaling levels. This diagram defines allowable and forbidden cell-fate transitions as EGF or Notch levels change, and explains surprising observations previously attributed to context-dependent action of these signals. The diagram also reveals the existence of special points at which minor changes in signal levels lead to strong epistatic interactions between EGF and Notch. Our model correctly predicts experiments near these points and suggests specific timed perturbations in signals that can lead to additional unexpected outcomes.

DOI: https://doi.org/10.7554/eLife.30743.001

**\*For correspondence:**
corson@lps.ens.fr (FC);
siggiae@mail.rockefeller.edu (EDS)

**Competing interests:** The authors declare that no competing interests exist.

## Introduction

Development is a dynamical process, so models that purport to be comprehensive must explicitly describe dynamics. Typically models report changes in protein levels and use them to predict phenotypic outcomes. However, the number of parameters involved makes implementation cumbersome and predictions non-intuitive. Classical embryology emerged in the absence of genetics and describes development in terms of overall cell and tissue phenotype. Such studies allow the inference of cell states that must exist even before any overt differentiation or morphogenesis is visible. For example, a cell is *competent* to respond to signals during a temporal window; it is *committed* or *specified* when those signals are no longer required, and *determined* when other signals cannot deviate it from its normal/assigned fate. Our aim here is to retain the conceptual clarity of classical embryology in models that made novel and quantitative predictions.

Developmental states admit an intuitive topographical representation, as proposed by Waddington and later formalized mathematically (*Waddington, 1957*; *Slack, 1991*). The development of a cell is conceived as a downhill path in a shifting landscape controlled by cell signaling. Between two outcomes, or valleys, is always a ridge, and cells poised on the ridge can descend into either valley with equal probability. Once pushed off a ridge, cell fates are determined irrespective of subsequent twists and turns of the valleys. The Waddington picture suggests that cell fate decisions can be separated from the complexity inherent in specification and morphogenesis, which inherently simplifies any model.

Vulval development in *Caenorhabditis elegans* (*Sternberg, 2005*) is an appealing setting in which to quantify Waddington's landscape metaphor. Here, six vulval precursor cells (VPCs), P3.p-P8.p, which are developmentally equivalent (P3.p is less competent and ignored in the model), receive an EGF signal from the anchor cell (AC), and interact through Notch signaling, to eventually assume

**eLife digest** At first, embryos are made up of identical cells. Then, as the embryo develops, these cells specialize into different types, such as heart and brain cells. Chemical signals sent and received by the cells are key to forming the right type of cell at the right time and place. The cellular machinery that produces and interprets these signals is exceedingly complex and difficult to understand.

In the 1950s, Conrad Waddington presented an alternative way of thinking about how an unspecialized cell progresses to one of many different fates. He suggested visualizing the developing cell as a ball rolling along a hilly landscape. As the ball travels, obstacles in its way guide it along particular paths. Eventually the ball comes to rest in a valley, with each valley in the landscape representing a different cell fate. Although this "landscape model" is an appealing metaphor for how signaling events guide cell specialization, it was not clear whether it could be put to productive use.

The egg-laying organ in the worm species *Caenorhabditis elegans* is called the vulva, and is often studied by researchers who want to learn more about how organs develop. The vulva develops from a small number of identical cells that adopt one of three possible cell fates. Two chemical signals, called epidermal growth factor (EGF) and Notch, control this specialization process.

Corson and Siggia have now constructed a simple landscape model that can reproduce the normal arrangement of cell types in the vulva. When adjusted to describe the effect of genetic mutations that affect either EGF or Notch, the model could predict the outcome of mutations that affect both signals at once. The twists and turns of cell paths in the landscape could also account for several non-intuitive cell fate outcomes that had been assumed to result from subtle regulation of EGF and Notch signals.

Landscape models should be easy to apply to other developing tissues and organs. By providing an intuitive picture of how signals shape cellular decisions, the models could help researchers to learn how to control cell and tissue development. This could lead to new treatments to repair or replace failing organs, making regenerative medicine a reality.

DOI: https://doi.org/10.7554/eLife.30743.002

one of three different terminal fates (*Sternberg and Horvitz, 1989*). Cells P6.p and P5/7.p assume the 1° and 2° fates, respectively, and after three divisions form the vulva. The remaining cells, P3/4.p and P8.p, are assigned fate 3°. They divide once and fuse with the hypodermis.

These basic facts suggested a model in which each cell travels in a landscape with three valleys (fates) that we represent in two dimensions to allow EGF and Notch to tilt the landscape independently as development proceeds (*Corson and Siggia, 2012*). The movement of a cell in the landscape depends on parameters that quantify the influence of each signal on the direction of motion in the landscape. Values for these parameters were obtained from known terminal VPC fate patterns of animals defective in the two signaling pathways, as well as from limited time-specific perturbations (ablation of the AC at different stages) (*Greenwald et al., 1983*; *Ferguson and Horvitz, 1985*; *Sternberg, 1988*; *Sundaram and Greenwald, 1993*; *Koga and Ohshima, 1995*; *Simske et al., 1995*; *Shaye and Greenwald, 2002*; *Félix, 2007*; *Milloz et al., 2008*; *Komatsu et al., 2008*; *Hoyos et al., 2011*). Partially penetrant phenotypes are ideal for parameter fitting as they define the locations of ridges. From this data alone, focusing on the competence period (*Ambros, 1999*; *Wang and Sternberg, 1999*), we built a quantitative model for how EGF and Notch signals control fates, without considering the underlying complex genetic networks (*Corson and Siggia, 2012*). Our model has no fitting parameters that would couple the EGF and Notch pathways, implying that that if we fit two alleles, one in each pathway, then the outcome of the cross is defined with no additional freedom. Still, the model is sufficient to explain experiments showing epistasis, including non-intuitive interactions that were previously attributed to a context-dependent action of the signals, for example low EGF can promote 2° fate acquisition, or biochemical interactions between the pathways in a single cell. The model can predict context-dependent signals since it applies a nonlinear function to a linear combination of the two vectors representing the pathways.

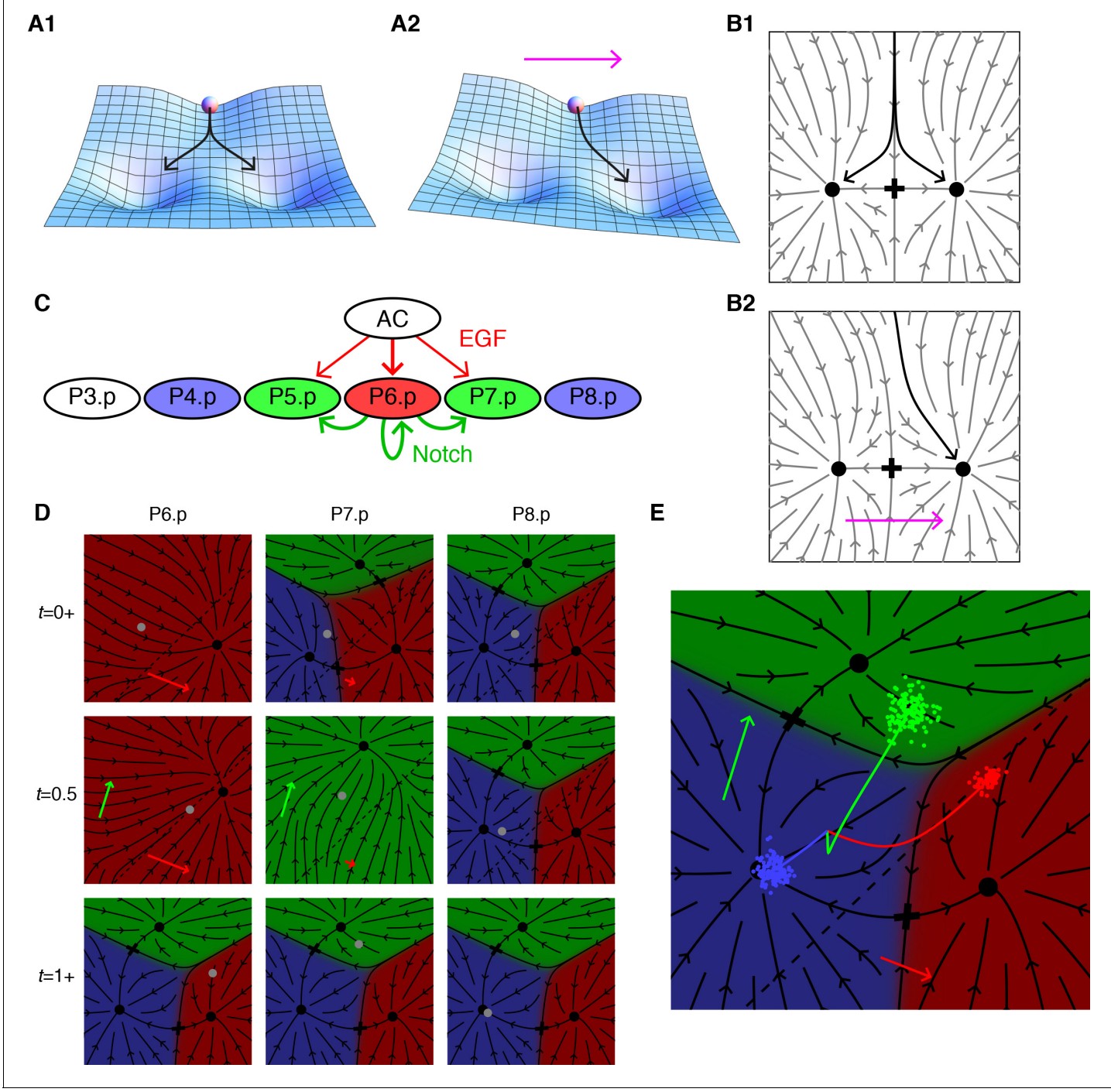

**Figure 1.** Model dynamics and the vulval patterning network. (A) Topography with two valleys separated by a ridge. A ball positioned exactly on the ridge will fall with equal probability into either valley. Tilting the topography as suggested by the arrow favors the right valley. (B) The same dynamics as A now represented by arrows showing the direction of movement at each point of the fate plane. The terminal points are shown as dots and the lowest point on the ridge that divides the flows is the saddle point denoted by a cross. (C) The vulva forms from a row of six 6 VPCs, P3.p-P8.p (P3.p differs in competence from other VPCs and is omitted from our model). Fate specification involves EGF signaling (red arrows) from the anchor cell (AC) and lateral Notch signaling among VPCs (green arrows). EGF induces P6.p to the 1° fate (red), while Notch signaling is required for the induction of the 2° fate in P5/7.p (green); this may be facilitated by low levels of EGF from the AC. P4/8.p are uninduced and adopt the default 3° fate (blue). Experiments in which an isolated VPC adopts the 2° fate suggest that the Notch receptor may also respond to autocrine signaling (*Sternberg and Horvitz, 1989*). (D) Snapshots showing the principal features of our graphical representation of the vulval fate plane following B. The VPC is represented by a grey dot, and all points that limit to a single fate are colored as in C, the arrows show the strength of the EGF (red) and Notch (green)

*Figure 1 continued on next page*

*Figure 1 continued*

signals, and cells positioned below the dashed line express Notch ligands. At the beginning of competence (*t* = 0+ signal reception on), all cells in an equivalence group start at the same position, but P6.p receives a strong EGF signal from the AC which makes 1° the only possible fate. Since P6.p is far from the dashed line there is no Notch signaling to P5/7.p. At mid-competence, P6.p reaches the dashed line and produces Notch ligands. This yields autocrine signaling in P6.p (green arrow) and paracrine signaling in P5/7.p that eliminates all fates except 2°. At the end of competence (t = 1+ signal reception off) the fate plane reverts to its configuration in the absence of signaling, which is virtually identical to that for P4/8.p at earlier times since it receives no signals under WT conditions. However, the induced cells find themselves in the red or green regions and progress toward the 1° or 2° fates. *Video 1* shows the full dynamics from pre to post competence. (E) A more compact representation of model dynamics, used in subsequent figures, overlays the average trajectories of P6-8.p, colored according to their WT fate, on the fate plane in the absence of signaling following the same notations as D. The clouds of colored dots represent outcomes for individual cells at the end of competence, after allowing for developmental noise. In other genetic backgrounds, this is the source of partial penetrance. Under the WT conditions shown here, each cloud is well inside a single territory, corresponding to an invariant fate pattern.

DOI: https://doi.org/10.7554/eLife.30743.003

The following figure supplement is available for figure 1:

**Figure supplement 1.** Dynamics and phase diagram of our original model (***Corson and Siggia, 2012***).

DOI: https://doi.org/10.7554/eLife.30743.004

Recent work (***Barkoulas et al., 2013***) quantified EGF levels in a series of perturbation lines and examined the cell fate outcomes in multiple combinations with Notch perturbation lines. The new data allows further quantitative refinement of the model, and an extensive test of its predictions. Using the refined model, we now generate a phase diagram that displays the terminal vulval cell fate pattern as a function of the EGF and Notch signal strengths. The diagram predicts the existence of special transition points where epistasis is particularly evident. These special points organize the entire phase diagram and define allowed and forbidden phenotypic transitions in response to continuous changes in signal strength. Verification of these transition points is an important goal of future experiments. Our data demonstrate that taking a more abstract, yet mathematically rigorous, geometric approach can reveal underlying principles of development, and can provide immediate quantitative predictions more difficult to obtain through kinetic modeling of the complex underlying gene and protein networks.

## Results

### A geometric model for *C. elegans* vulval development

To quantify the Waddington topography yet retain its intuitive appeal requires a *geometrical* representation of dynamics that we illustrate in two dimensions, *Figure 1A,B*. The axes of this so-called *fate plane* are abstract and only acquire meaning when we fit to actual data. Dynamics is represented in the fate plane by arrows (a *flow*) defining the velocity of a point representing the state of a cell. The developmental history of a cell is defined by tracing its trajectory in the fate plane. The very bottom of a valley is defined by a point with only inward pointing arrows. The low point on a ridge separating two valleys, a so-called *saddle point*, has two inward directions along the ridge and two outward sectors representing flow into the valleys.

Turning to *C. elegans* vulval patterning, we adopt a two-dimensional fate plane to accommodate the two signaling pathways, *Figure 1C* (***Corson and Siggia, 2012***). The VPCs initially form an equivalence group, thus the same model suffices for each cell, and the three possible fates require three distinct points at which cell trajectories stably terminate. Each of the VPCs starts from the same location at the beginning of competence, but their fate planes distort in different ways in response to EGF and Notch signals, yielding divergent trajectories (*Figure 1D* and *Video 1*). The VPCs receive graded EGF levels that are constant in time, consistent with

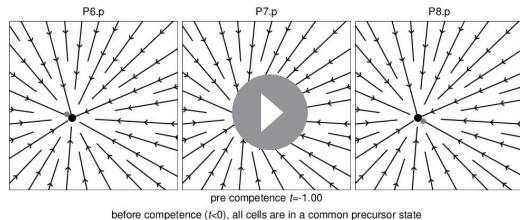

**Video 1.** Model dynamics of P6-8.p before, during and after the competence period, showing how the secondary Notch signals depend on the state of P6.p (cf. *Figure 1*).

DOI: https://doi.org/10.7554/eLife.30743.005

**Table 1.** Experimental data used to fit the model (training set).

Animals heterozygous for null mutations in *lin-3* or *lin-12* are described in the model by dividing the signal by two ('half dose'). The EGF level for one overexpression line (JU1107) was set according to the measured *lin-3* mRNA level. For other perturbations, the change in signal is included as a fitting parameter, cf. Table S2. * Notch null mutations result in the formation of two anchor cells, thus the effect of suppressing Notch signaling between VPCs it not obvious; however, several lines of evidence suggest that it yields the pattern 33133.

| Experiment | VPC fates (fate or % 1°, 2°, 3°) | | | | | References |
| --- | --- | --- | --- | --- | --- | --- |
| | P4.p | P5.p | P6.p | P7.p | P8.p | |
| Wild type | 3° | 2° | 1° | 2° | 3° | |
| **EGF/Notch mutants** | | | | | | |
| *let-23* mosaic (no EGF receptors in P5/7 .p) | wild type | | | | | (*Koga and Ohshima, 1995*; *Simske et al., 1995*) |
| Half dose of EGF (*lin-3/+*) | wild type | | | | | (*Ferguson and Horvitz, 1985*) |
| Half dose of Notch (*lin-12/+*) | wild type | | | | | (*Greenwald et al., 1983*) |
| No Notch signaling * | 3° | 3° | 1° | 3° | 3° | (*Sundaram and Greenwald, 1993*; *Shaye and Greenwald, 2002*; *Komatsu et al., 2008*) |
| Notch null mutant, 2 ACs * (described as 2 × WT EGF) | 3° | 1° | 1° | 1° | 3° | (*Greenwald et al., 1983*) |
| *lin-15* mutant (ectopic EGF; fit to ~0.4 × WT) | alternating 1° and 2° fates (with or without AC) | | | | | (*Sternberg, 1988*) |
| **EGF overexpression** | | | | | | |
| JU1107 2.75 × WT EGF (based on measured mRNA level) | 2, 16, 82 | 22, 78, 0 | 100, 0, 0 | 16, 84, 0 | 3, 13, 84 | (*Barkoulas et al., 2013*) |
| JU1100 (mRNA level not determined; level fit) | 24, 54, 21 | 54, 46, 0 | 96, 4, 0 | 37, 63, 0 | 12, 39, 49 | (*Hoyos et al., 2011*) |
| **AC ablation at successive developmental stages** (mapped to evenly spaced time points in the model) | | | | | | (*Milloz et al., 2008*) |
| L2 lethargus (*t* = 0.2) | | 3° | 3° | 3° | | |
| early L3 (*t* = 0.32) | | 3, 24, 74 | 18, 19, 64 | 0, 20, 80 | | |
| DU divided (*t* = 0.44) | | 0, 59, 41 | 31, 37, 32 | 0, 53, 47 | | |
| VU divided (*t* = 0.56) | | 0, 95, 5 | 53, 48, 0 | 8, 86, 6 | | |
| 3° divided (*t* = 0.68) | | 3, 98, 0 | 65, 35, 0 | 0, 100, 0 | | |
| two-cell stage (*t* = 0.8) | | 1, 99, 0 | 93, 7, 0 | 1, 99, 0 | | |
| **Epistasis between EGF/Notch** | | | | | | (*Barkoulas et al., 2013*) |
| CB1417 (*lin-3(e1417)* EGF hypomorph; level fit) | 0, 0, 100 | 0, 10, 90 | 54, 0, 46 | 0, 10, 90 | 0, 0, 100 | |
| JU2064 (mild ectopic Notch activity; level fit) | 0, 1, 99 | 0, 100, 0 | 100, 0, 0 | 0, 100, 0 | 1, 2, 97 | |
| JU2095 (mild ectopic Notch activity in the *lin-3(e1417)* background) | 0, 0, 100 | 1, 14, 86 | 72, 1, 28 | 0, 16, 84 | 0, 2, 98 | |

DOI: https://doi.org/10.7554/eLife.30743.006

observed mRNA levels in the AC (*Barkoulas et al., 2013*). Notch ligand production, on the other hand, is turned on in cells approaching the 1° fate (see the response in P7.p at *t* = 0.5 after P6.p has reached the dotted line in *Figure 1D*). The model also incorporates fluctuations to account for partially penetrant phenotypes. The states of a VPC for a group of animals is depicted as a cloud of points traveling in the fate plane (*Figure 1E*). With wild-type parameters, each of these clouds lies well within a valley following competence, corresponding to an invariant fate pattern.

**Table 2.** Model parameters.
The mean value and standard deviation within the parameter ensemble (where applicable) are indicated for each parameter.

| Parameter | Description | Main model | Parameter values from *Corson and Siggia (2012)* | Model with EGF/Notch coupling (*Figure 7*) | Alternate models of *Figure 2—figure supplements 3* and *4* |
|---|---|---|---|---|---|
| **Model parameters** | | | | | |
| $\vec{m}_0$ | Bias towards the default 3° fate | 0.47 ± 0.03 | 0.39 ± 0.03 | 0.51 ± 0.03 | 0.5 |
| $\angle\,\vec{m}_0$ | | 210 (fixed) | 210 (fixed) | 210 (fixed) | 210 |
| $\|\vec{m}_1\|$ | Response to EGF* | 3.87 ± 0.52 | 4.60 ± 0.98 | 3.86 ± 0.52 | 6 |
| $\angle\,\vec{m}_1$ | | −21 ± 8 | −30 (fixed) | −9 ± 6 | −30 |
| $\|\vec{m}_2\|$ | Response to Notch* | 6.25 ± 1.28 | 5.97 ± 1.07 | 7.72 ± 1.31 | 6 |
| $\angle\,\vec{m}_2$ | | 73 ± 3 | 90 (fixed) | 73 ± 5 | 90 |
| $\frac{1}{\tau}$ | Characteristic time scale of dynamics | 2.02 ± 0.16 | 2.18 ± 0.30 | 2.23 ± 0.25 | 2 |
| $\sqrt{2D}$ | Noise level | 0.12 ± 0.02 | 0.12 ± 0.03 | 0.14 ± 0.02 | 0.14 |
| $\gamma$ | Shape of EGF gradient | 0.23 ± 0.02 | 0.16 ± 0.03 | 0.22 ± 0.02 | 0.2 |
| $\alpha$ | Relative strength of autocrine signaling | 1.08 ± 0.11 | 1.14 ± 0.17 | 1.24 ± 0.22 | 1/2 (*Figure 2—figure supplement 3*) or 0 (*Figure 2—figure supplement 4*) |
| $n_0$ | Threshold for lateral signaling | −1.20 ± 0.09 | −1.23 ± 0.10 | −1.29 ± 0.11 | 0.25 |
| $\angle\,\vec{n}_1{}^†$ | | −46 ± 3 | −48 ± 3 | −46 ± 3 | −45 |
| **Signal levels for experiments in the training set** | | | | | |
| | Ectopic EGF level in the *lin-15* mutant | 0.42 ± 0.11 | 0.55 ± 0.25 | 0.51 ± 0.20 | N/A |
| | EGF level in the JU1100 overexpression line | 4.18 ± 0.49 | 7.39 ± 1.36 | 4.84 ± 0.67 | N/A |
| | EGF level in the *lin-3 (e1417)* hypomorph | 0.28 ± 0.03 | N/A | 0.28 ± 0.03 | N/A |
| | Ectopic Notch activity in the JU2064 line | 0.05 ± 0.03 | N/A | 0.04 ± 0.02 | N/A |

*While the vectors $\vec{m}_1$ and $\vec{m}_2$ are parameterized by their Cartesian coordinates, with Gaussian priors, the mean ±SD of their norm and orientation are shown for convenience. Angles (symbolized by $\angle$) are relative to the horizontal axis of the fate plane and in degrees.

†As in (*Corson and Siggia, 2012*), the sharpness of the threshold for lateral signaling was set to a default value, $\|\vec{n}_1\| = 3$, so that its orientation alone is fit; a similar value for the norm of $\vec{n}_1$ is recovered when it is not imposed in the fit.

DOI: https://doi.org/10.7554/eLife.30743.007

New data from *Barkoulas et al. (2013)* provide VPC fates for a set of lines with defined EGF levels crossed into strains with perturbed Notch levels to generate a grid of data. This invites comparison with the outcome of the model as signal levels are continuously varied that is, a phase diagram. While our previous model is consistent with experimental outcomes, we can use some of the new data to more finely estimate its parameters. The new fit is fully described under Methods (see also *Tables 1* and *2*, *Figure 1—figure supplement 1*) and its implications explored here.

## A phase diagram predicts novel, quantitative, features of cell fate determination

We ran our model for a range of Notch and EGF signaling levels, and determined the collective fate, or *pattern*, of the VPCs for each condition. The *phase diagram* in *Figure 2* displays the outcome of our simulations and makes some surprising facts intuitive. Its power derives from the empirical observation that VPC fates, hence vulval patterns, are generally discrete and retain their identity under mutations in the signaling pathways. We highlight four notable features that are common to any system with two signals regulating three fates.

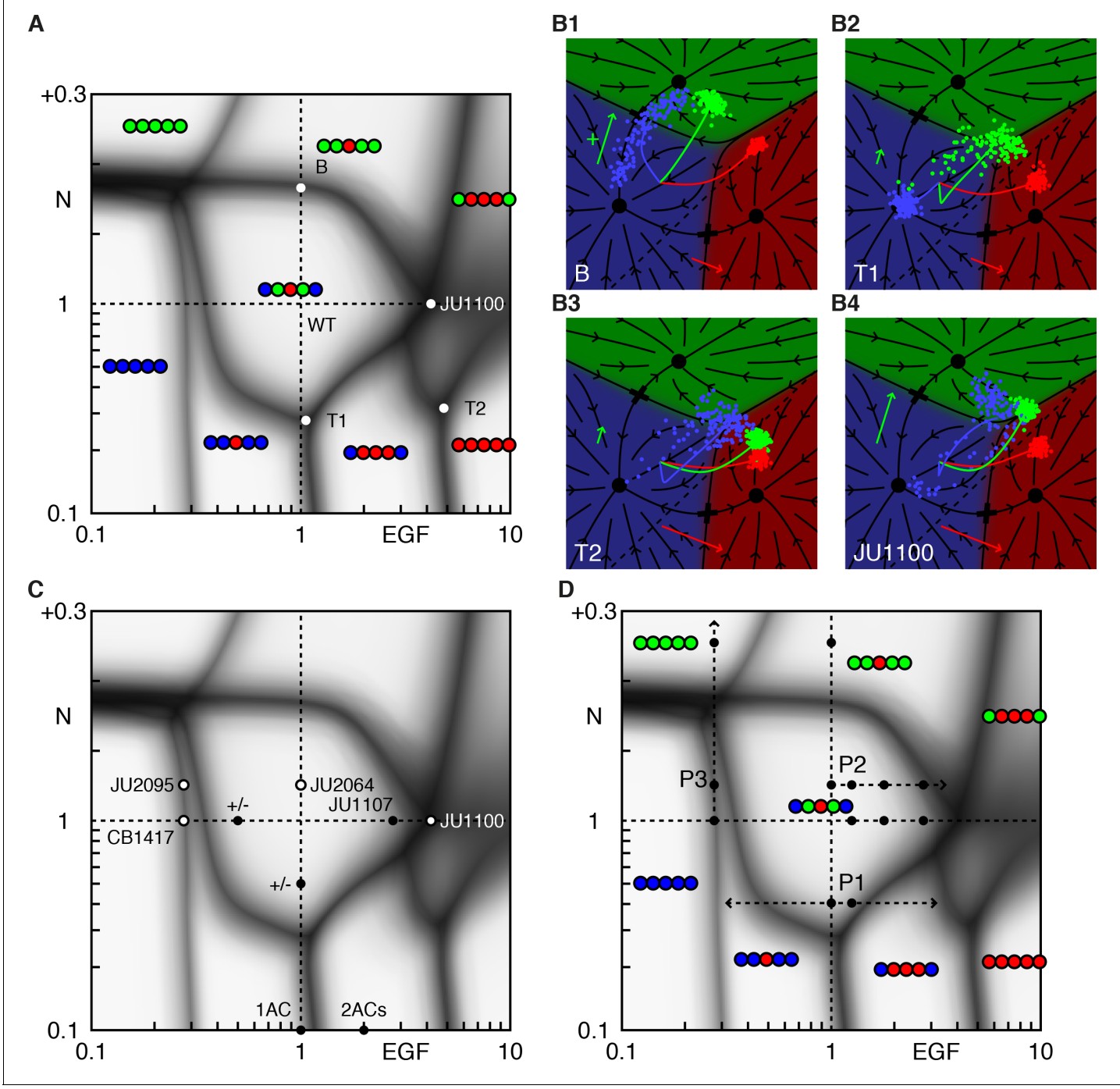

**Figure 2.** Phase diagram of the model. (**A**) Phase diagram of the model showing the vulval pattern as a function of EGF and N(otch) levels normalized to the WT value of one. The horizontal axis represents fold-change in EGF pathway activity, while the vertical is defined to follow (***Barkoulas et al., 2013***). Notch levels below one are fold changes induced by RNAi, while larger values represent a uniform level of NICD added to the WT level in all cells (in units of the maximum signal produced by a cell). The shaded regions represent zones of partial penetrance marking the transition between two or more discrete fates. Three boundaries meet in *triple points*, e.g., T1/2. The labeled points are elaborated in B. (**B**) The fate plane depiction for transition points marked in A. The boundary point B corresponds to mild ectopic Notch activity (symbolized by a green '+'), resulting in partial conversion of P4/8.p to the 2° fate. The triple points T1 and T2 are two dosage combinations such that one pair of cells (P5/7.p and P4/8.p, respectively) can yield all three fates. JU1100 is an EGF over expression line (fit to ~4 × WT; longer red arrow), in which both P4/8.p and P5/7.p are partially transformed. Other symbols follow ***Figure 1E***. (**C**) The data used to fit the model that admits representation in the phase diagram is shown as solid dots when both the phenotype and signal levels are used, and open circles when only the phenotype was assumed. The +/- denote a half dose of respective signal. (**D**) The data from (***Barkoulas et al., 2013***) (dots) overlaid on the phase diagram. The paths P1-3 are discussed in subsequent figures.

*Figure 2 continued on next page*

*Figure 2 continued*

DOI: https://doi.org/10.7554/eLife.30743.008

The following figure supplements are available for figure 2:

**Figure supplement 1.** Phase diagram with a different fate plane topology.

DOI: https://doi.org/10.7554/eLife.30743.009

**Figure supplement 2.** Phase diagrams for a deterministic model.

DOI: https://doi.org/10.7554/eLife.30743.010

**Figure supplement 3.** Alternative dynamics with enhanced production of Notch ligands.

DOI: https://doi.org/10.7554/eLife.30743.011

**Figure supplement 4.** Alternative dynamics following *Figure 2—figure supplement 3* but without autocrine Notch signaling.

DOI: https://doi.org/10.7554/eLife.30743.012

**Figure supplement 5.** Response to EGF/Notch perturbations.

DOI: https://doi.org/10.7554/eLife.30743.013

1. There exist continuous domains in the Notch-EGF plane where the fate pattern is unique. The wild-type domain reflects the range of EGF and Notch perturbations that still lead to normal development, assigning a quantitative measure to the 'robustness' of the system.
2. Domains are separated by fuzzy lines, reflecting partial penetrance, that mark transitions between patterns. Transitions from one pattern to another are generally restricted to fate changes of a single VPC (or a symmetric pair). As shown in *Figure 2*, at transition points, the cloud of cell states is stretched out across the boundary between two fates, corresponding to a variable outcome (*Figure 2B1*).
3. Domain boundaries are not parallel to the coordinate axes. Thus, cell fate is not uniquely specified by the levels of either EGF or Notch. Rather, a combination of these signals defines the terminally differentiated state.
4. Boundary lines typically meet at three-way junctions, which we term triple points to follow the thermodynamic analogy. Near these points, small changes in EGF, Notch, or both can produce very different fate patterns, providing a strong test of our model. At a triple point, a single VPC can assume one of three fates (*Figures 2B2* and *2B3*).

Triple points evidently require that any pair of fates have a coexistence boundary in the fate plane as is evident in *Figure 1E*. A cloud of points induced by noise that lands near the point where the three phases meet, will populate all three fates. In the limit of a pure graded induction model for the vulva, *Figure 2—figure supplement 1*, the 2° fated domain sits between the 1° and 3° fates. The fate plane is effectively one-dimensional (flow onto a line with two saddles separating three fates) and triple points cannot occur. This configuration is ruled out by the anchor cell ablation data (*Corson and Siggia, 2012*; *Félix, 2007*; *Milloz et al., 2008*). As the ablation time increases, P6.p transits from the 3° fate to equal fractions of 1° and 2°, implying boundaries between all three domains.

Aspects of the phase diagram are specific to vulva but obvious without a model. Under low Notch, the pattern transitions from 33333, to 33133, 31113, and 111111 as EGF levels increase and become sufficient to induce more distal cells. Under low EGF, strong ectopic Notch activity yields all 2° fates. Some quantitative features are straightforward consequences of the experimental data, *Figure 2C*. EGF and Notch null mutations are recessive, that is, a half-dosage is sufficient for patterning (+/− labels); this and an EGF overexpression line with known EGF levels (JU1107 label; see *Table 3* for a list of the experimental lines referred in this study) delimit the central wild-type region. Similarly, data on animals with impaired Notch function and 1 or 2 ACs position the boundary between 33133 and 31113.

Elsewhere, the structure of the phase diagram is less intuitive and requires explanation. Around $0.3 \times$ WT EGF and WT$+ 0.2$ Notch, small changes in EGF and/or Notch lead to five different fate patterns. Removing the molecular noise, *Figure 2—figure supplement 2B*, reveals that the triple point where P6.p assumes all three fates lies very close to where the other VPCs convert to fate 2°. This degeneracy is caused by the simultaneous conversion of all five VPCs to 2° fates for EGF = 0.

A second confluence of multiple phases near the JU1100 point ($\sim 4 \times$ WT EGF), as well as the very narrow domain of 33133 for Notch >1, both are due to how data positions the dashed line in *Figure 1E* for the onset of Notch ligand production: near to the boundary between fates 1° and 2°. The JU1100 fate proportions (P4/8.p partially converted to 1° and 2° fates, and P5/7.p partially

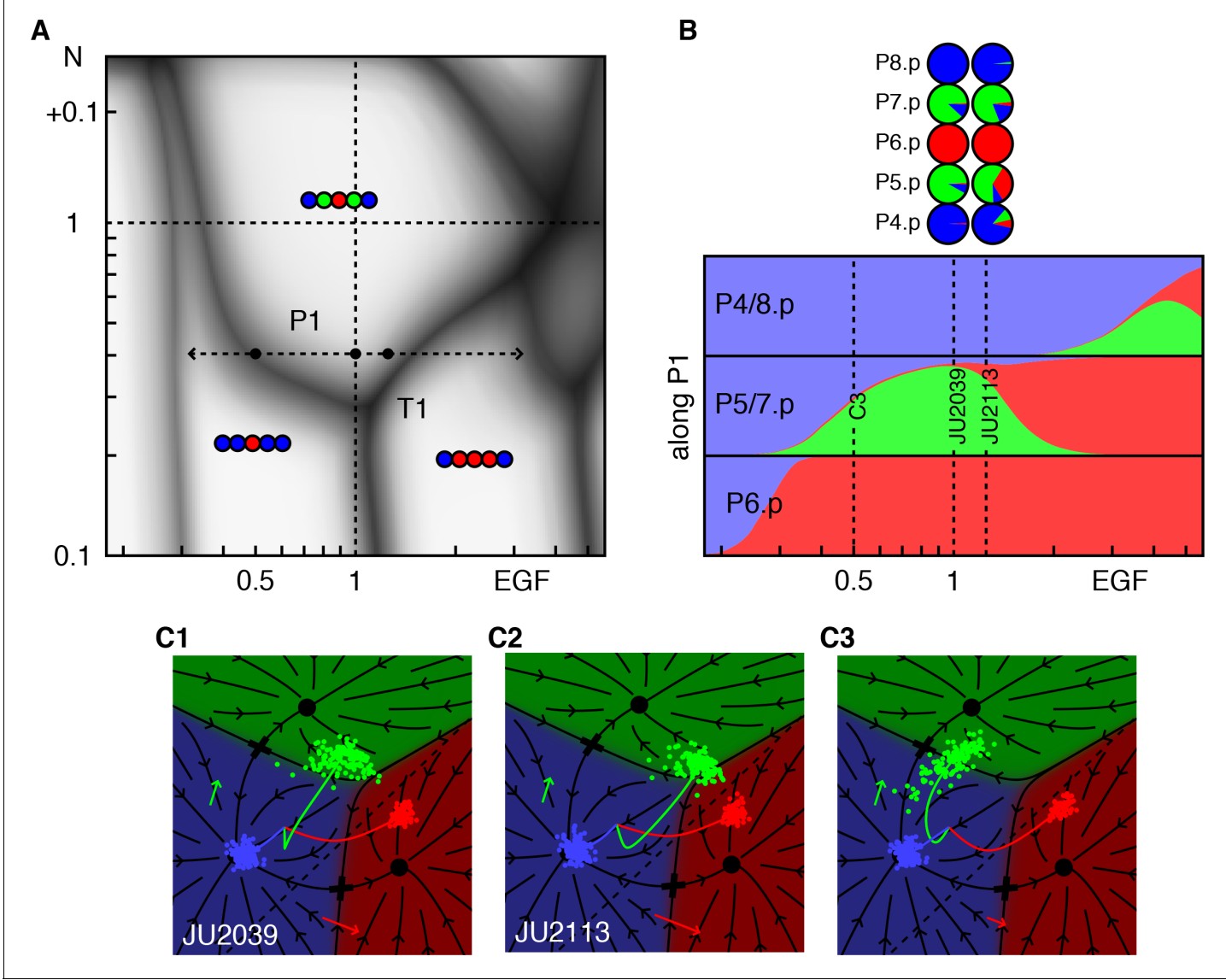

**Figure 3.** Response to EGF perturbations under reduced Notch. (**A**) Blowup of the phase diagram of *Figure 2* around the triple point T1, where P5/7.p assume three fates. The double-headed arrow (path P1) represents variations in EGF dosage in a background with reduced Notch (*lin-12* RNAi (*Barkoulas et al., 2013*) fit to ~0.4 × WT), which lies just above the lower corner of the wild-type domain. The dots on P1 are examined in panels B,C. (**B**) Cell fate proportions as a function of EGF dosage from the model and data (pie plots). A 25% increase in EGF already yields a significant fraction of adjacent 1° fates. The pronounced anterior-posterior asymmetry in the data is not seen in the model and may represent enhanced sensitivity to anterior displacements of the AC (*Sundaram and Greenwald, 1993*). Moving along P1 to 0.5 × WT EGF, we predict loss of 2° fates in P5/7.p. (**C**) Fate plane for different EGF dosages (WT, 1.25 × WT, and 0.5 × WT). In C1 the fit places the *lin-12* RNAi condition just above the triple point T1 of the phase diagram, corresponding to P5/7.p (green dots) just above the center of the fate plane where the three domains meet. In C2, a very minor increase in EGF is sufficient to convert a fraction of the cells to the 1° fate. If instead reduced Notch is combined with a silent reduction in EGF (half dose), a substantial loss of 2° fates is predicted in C3 (*Table 4*).

DOI: https://doi.org/10.7554/eLife.30743.014

converted to 1° fate) were used to fit and are represented as a triple point (P4/8.p assume all three fates) nearly coincident with the boundary between WT and 31113, *Figure 2—figure supplement 2A*. Similarly, the phenotype of the CB1417 line implies that loss of the 1° and 2° fates occurs around the same EGF level. If we shift the signaling threshold away from the 1° domain, these fate confluences disappear and different patterns become accessible, *Figure 2—figure supplements 3* and *4*. (It is perfectly possible to observe with Notch = 1 the patterns 32223, 32123, and 22122 as EGF

increases from below one to above, while retaining a large domain for the WT pattern signifying its robustness).

Importantly, our model makes precise predictions about the activity levels of the EGF and Notch pathways at interesting points in the phase diagram. *Figure 2D* overlays the new experiments from (*Barkoulas et al., 2013*) on our phase diagram. In all but one case (JU1100), the EGF levels were measured while Notch levels are not and their values constitute a prediction. For the lowest expressing EGF hypomorph, *lin-3(e1417)*, (denoted as CB1417 in (*Barkoulas et al., 2013*)), our model could not readily accommodate the very low EGF mRNA level in (*Barkoulas et al., 2013*) and the reported phenotype (*Table 1*). If, however, we use the model to predict the EGF level from the phenotype (*Table 2*), it agrees quite well with another measurement of the same allele reported in (*Saffer et al., 2011*) (*Table 3*). Elsewhere the correspondence between EGF level and VPC fates agrees well with experiments (*Figure 2—figure supplement 5B*).

Taken together, these examples reveal unexpected structure in the phase diagram that could not be gleaned from the available data without a model.

## Triple points reveal abrupt fate changes

The phase diagram in *Figure 2A* predicts that at certain Notch and EGF signal strengths, strong genetic interactions can be observed, for example the triple point T1, where P5/7.p adopt equal proportions of all three fates. Our fit places this point close to WT EGF levels, and 0.3 × WT Notch activity, thus a moderate reduction in Notch creates a novel type of 'sensitized background' in which modest increases or decreases in EGF both lead to loss of 2° fates (see path P1 in *Figure 3A*).

This prediction is born out by experiment (*Barkoulas et al., 2013*). Whereas animals subjected to RNAi against *lin-12*/Notch (fit as 0.4 × WT Notch) show occasional conversion of P5/7.p to the 3° fate, elevating EGF to only 1.25 × WT results in a substantial fraction of P5/7.p cells adopting a 1° fate, *Figure 3B–C* and *Table 4*. We predict that in the same RNAi background, removing one EGF copy (50% EGF activity) would result in substantial conversion of P5/7.p to the 3° fates. The different outcomes of Notch reduction under different EGF levels could be taken to suggest a switch between two functions of Notch, induction of the 2° fate and inhibition of the 1° fate (*Barkoulas et al., 2013*). However, our model allows an apparently simpler explanation: EGF and Notch function

**Table 3.** Experimental lines from (*Barkoulas et al., 2013*), and one line from (*Hoyos et al., 2011*), referred in this study. Levels used to model the lines were based on measurements of EGF mRNA levels from (*Barkoulas et al., 2013*) or fit, as indicated. For combined perturbations, we indicate the corresponding single perturbations lines.

| | |
|---|---|
| **EGF perturbation lines** | |
| CB1417 | *lin-3(e1417)* hypomorph; level included as fitting parameter ~0.28 × WT (measured as ~0.06 × WT in (*Barkoulas et al., 2013*) and ~0.22 × WT in [*Saffer et al., 2011*]) |
| JU2036 | measured as ~ 1.25 × WT |
| JU2035 | measured as ~ 1.79 × WT |
| JU1107 | measured as ~ 2.75 × WT |
| JU1100 (*Hoyos et al., 2011*) | level included as fitting parameter ~4.2 × WT (not measured) |
| **Notch perturbation lines** | |
| JU2039 | Pn.p cell-specific RNAi; level fit to phenotype strength ~0.4 × WT |
| JU2064 | ectopic Notch activity; level included as fitting parameter ~WT + 0.05 |
| JU2060 | ectopic Notch activity; level fit to yield near-complete P4/8.p conversion ~ WT + 0.26 |
| **Combined perturbations** | |
| JU2113 | JU2036 × JU2039 |
| JU2091 | JU2036 × JU2064 |
| JU2089 | JU2035 × JU2064 |
| JU2092 | JU1107 × JU2064 |
| JU2095 | CB1417 × JU2064 |

DOI: https://doi.org/10.7554/eLife.30743.015

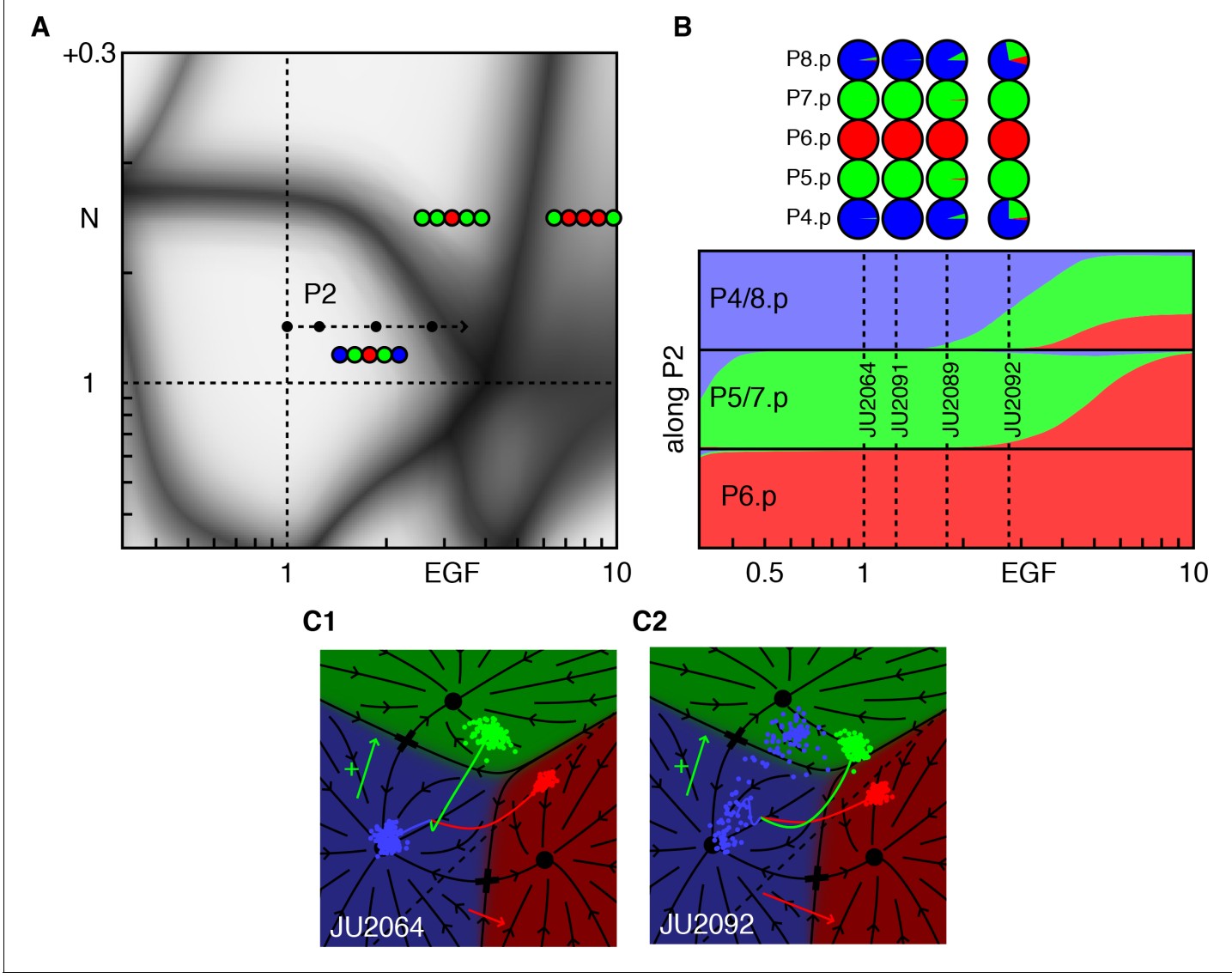

**Figure 4.** Epistasis between excess EGF and Notch. (**A**) Blowup of the phase diagram of *Figure 2* centered on the region with excess EGF and Notch. The arrow (P2 ) represents gradual increases in EGF in a background with mild ectopic Notch activity (fit to ~WT + 0.05). (**B**) Fate proportions as a function of EGF dosage along path P2, in the model and experiments for the points in A (*Table 4*). (**C**) Cell trajectories in the fate plane for the JU2064 line (ectopic Notch and WT EGF) and the JU2092 line, a cross with the strongest EGF overexpression, explaining how excess EGF can produce more 2° fates in P4/8.p (text). See *Video 2*.

DOI: https://doi.org/10.7554/eLife.30743.017

independently, and because the path P1 cuts through a corner of the WT domain, changes up or down in EGF can result in conversion of 2° to 1° or 3° fates.

## Transient signaling explains how EGF favors adjacent 2°

The confluence of several patterns at the end of the path P2 in *Figure 4A* and the new data from (*Barkoulas et al., 2013*) allow a test of our assertion that the phenotype of a genetic cross can be predicted by fitting only the single mutants. The strain JU2092 (*Figure 4B,C2*, *Table 4*) combines modest ectopic Notch activity (whose level we fit from an independent experiment, *Figure 2C*), with a defined level of EGF overexpression. In these animals, P4/8.p cells are partially converted to the 2° fate while leaving P5-7.p unaffected. This is quite surprising, as increasing EGF leads to a 2° fate without an adjacent 1° fate cell. Our model provides a simple, yet unexpected, explanation

**Table 4.** Comparison between model predictions and experimental outcomes from (*Barkoulas et al., 2013*).
The model prediction (mean ±SD within the parameter ensemble) is displayed below the experimental outcome.

| Experiment | VPC fates (% 1°, 2°, 3°) | | | | |
| --- | --- | --- | --- | --- | --- |
| | P4.p | P5.p | P6.p | P7.p | P8.p |
| **Excess EGF × reduced Notch (*Figure 3*)** The Notch level is our fit for Notch RNAi in the JU2039 line ~ 0.4 × WT. The EGF level is based on the mRNA level measured in the JU2036 EGF perturbation line (*Barkoulas et al., 2013*). | | | | | |
| JU2039 (WT EGF) | 1, 0, 99 | 1, 91, 8 | 100, 0, 0 | 1, 87, 11 | 0, 0, 100 |
| | 0 ± 0, 0 ± 0, 99 ± 0 | 3 ± 1, 88 ± 1, 8 ± 1 | 99 ± 0, 0 ± 0, 1 ± 0 | 3 ± 1, 89 ± 1, 8 ± 1 | 0 ± 0, 0 ± 0, 99 ± 0 |
| JU2113 (1.25 × WT EGF) | 7, 10, 83 | 32, 60, 8 | 100, 0, 0 | 4, 79, 17 | 0, 2, 98 |
| | 0 ± 0, 0 ± 0, 99 ± 0 | 16 ± 8, 76 ± 9, 9 ± 2 | 99 ± 0, 0 ± 0, 1 ± 0 | 16 ± 9, 75 ± 10, 9 ± 2 | 0 ± 0, 0 ± 0, 99 ± 0 |
| **Excess EGF × ectopic Notch activity (*Figure 4*)** Increasing EGF levels in a background with mild ectopic Notch activity. The level of ectopic Notch activity is our fit for the JU2064 line ~ WT + 0.05. EGF levels are based on measured mRNA levels in the EGF perturbation lines JU2036, JU2035, and JU1107 (*Barkoulas et al., 2013*). | | | | | |
| JU2091 (1.25 × WT EGF) | 0, 0, 100 | 0, 100, 0 | 100, 0, 0 | 0, 100, 0 | 0, 1, 99 |
| | 0 ± 0, 1 ± 1, 99 ± 1 | 1 ± 0, 99 ± 1, 1 ± 0 | 98 ± 0, 1 ± 0, 1 ± 0 | 1 ± 0, 99 ± 1, 1 ± 0 | 0 ± 0, 1 ± 1, 99 ± 1 |
| JU2089 (1.79 × WT EGF) | 0, 5, 96 | 2, 98, 0 | 100, 0, 0 | 2, 99, 0 | 1, 7, 92 |
| | 0 ± 0, 6 ± 5, 93 ± 5 | 2 ± 1, 97 ± 2, 1 ± 1 | 99 ± 0, 1 ± 0, 1 ± 0 | 2 ± 1, 97 ± 2, 1 ± 1 | 1 ± 0, 6 ± 5, 93 ± 6 |
| JU2092 (2.75 × WT EGF) | 3, 24, 74 | 0, 100, 0 | 100, 0, 0 | 0, 100, 0 | 8, 24, 68 |
| | 1 ± 1, 38 ± 14, 61 ± 14 | 9 ± 4, 88 ± 5, 3 ± 1 | 99 ± 0, 1 ± 0, 1 ± 0 | 8 ± 4, 88 ± 5, 3 ± 1 | 1 ± 1, 38 ± 14, 61 ± 14 |

DOI: https://doi.org/10.7554/eLife.30743.016

(*Figure 4C2*, *Video 2*). When EGF levels are sufficiently large, we predict that P5/7.p transit through the 1° fate domain on their way to the 2° fate, and thus transiently signal to P4/8.p. This transient signal synergizes with ectopic Notch activity to induce 2° fates in P4/8.p. This prediction depends critically on how we model Notch signaling in the fate plane (dashed line in *Figure 1E*), a fit obtained from different experiments. The same position of the line for Notch signaling is independently required to explain the outcome of AC ablation experiments (*Corson and Siggia, 2012*). Thus, our geometric model remarkably ties together disparate experimental facts.

## Cell intrinsic dynamics reveal how EGF/Notch can favor 2°/1° fates

Yet another observation affirming a context-dependent association of signals and fates is the observation that ectopic Notch activity favors the 1° fate in an EGF hypomorph (*Barkoulas et al., 2013*) (path P3 in *Figure 5A,B*). This fact appears in the phase diagram as a slight inclination of the boundary separating the 33333 territory from 33133, *Figure 5A*. This counterintuitive behavior of Notch is easily explicable from the trajectories in the fate plane of the model, *Figure 5C*. The EGF hypomorph leaves P6.p poised between fates 3° and 1° and stretched out along the line connecting the saddle or decision point to the associated fates. But the Notch signal has no reason to be parallel to the 'ridge' leading into the saddle and it in fact favors fate 1°, explaining the increased 1° fates with ectopic Notch. More generally as a function of increasing Notch, we predict a peak in the fraction of 1° fated P6.p followed by their conversion to entirely 2° fates for highest Notch as observed (*Barkoulas et al., 2013*), *Figure 5B*.

A converse experiment (*Zand et al., 2011*) found that mild EGF signaling could induce the 2° fate in a sensitized background of weak ectopic Notch that alone was not strong enough to induce appreciable 2° fates. The fate plane of the model, *Figure 5D*, shows that the sensitized state is poised near the saddle point between fates 2° and 3°. Again because the vector representing EGF is not parallel to the flow into the decision point, EGF can tip the balance and favors 2°. We could explain the enhanced fraction of 2° fates induced by EGF observed in *Zand et al. (2011)*, merely from the flow in the fate plane around the saddle point, with no special assumptions. By contrast, Ref. (*Zand et al., 2011*) related this outcome to a switch between different Ras effectors, suggesting (if we translate to our language) that the EGF vector, whose direction remains fixed in our model, rotated and pointed upward toward 2° and away from 1° for low ligand levels.

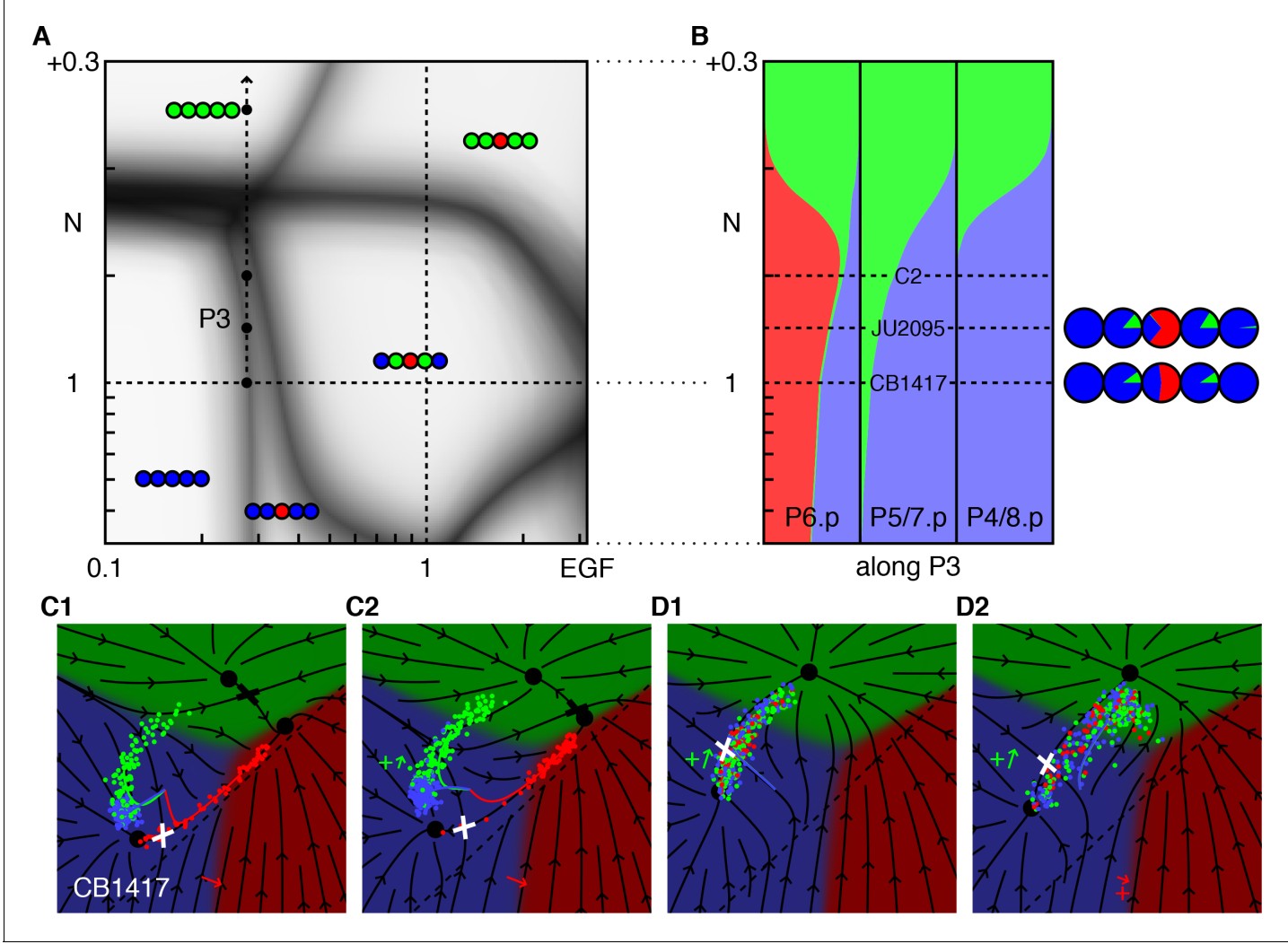

**Figure 5.** Synergy between low EGF and low ectopic Notch. (**A**) Blowup of the phase diagram of *Figure 2* showing the region with reduced EGF and excess Notch. The arrow (P3) represents gradual increases in ectopic Notch activity in the background of an EGF hypomorph, *lin-3(e1417)* or CB1417 in (*Barkoulas et al., 2013*) (fit to ~0.28 × WT). The boundary between the 33333 and 33133 domains makes a slight angle with the vertical, thus induction of P6.p to the 1° fate is partially rescued by Notch. (**B**) Fate proportions vs. Notch dosage in the model. The red bump in the left column reflects the partial rescue of P6.p induction by mild ectopic Notch activity. Under higher Notch doses, P6.p is converted to the 2° fate, as observed in a comparable experiment with EGF RNAi (*Barkoulas et al., 2013*). Correspondence with the data in (*Barkoulas et al., 2013*) is shown with the pie charts. An extra condition, C2, near maximum induction, was added to illustrate the model in the next panel. (**C**) Model trajectories for the EGF hypomorph, and the cross with a silent Notch gain-of-function line (N = WT + 0.1, the line C2 in B). In contrast to prior fate plane depictions, we show with arrows the flow experienced by P6.p during the competence window, including autocrine Notch signaling. Thus, the red trajectory follows the arrows. All cells begin at the same location at the beginning of competence (intersection of colored lines). The P6.p cells move toward the saddle or decision point (white cross) that separates the 1° and 2° fates, and then spread out along the line exiting that point. In C2, a small ectopic Notch signal pushes the P6.p points away from the decision point and more flow into the red 1° fate domain. (**D**) Enhancement of 2° induction by mild EGF activity (*Zand et al., 2011*). In this Notch sensitized background (N = WT + 0.16, D1), there is no AC or EGF signaling, and cells flow into the saddle point (white cross) separating the 3° and 2° fates and then diverge into two streams, using the same representation as in C. In D2 we add ectopic EGF activity [*lin-15(ts)*] (EGF = 0.16, +red arrow) which moves cells away from the line that flows into the saddle point. Thus, more cells are deflected toward the 2° fate by the flow around the saddle point. EGF never points toward the 2° fate, yet by acting early in the competence period can effect a large change in outcomes precisely because the background in D1 is poised near the saddle point and the flow around the cross rotates by 90°.

DOI: https://doi.org/10.7554/eLife.30743.019

## Dynamical perturbations are sensitive tests of the model

Time-dependent perturbations during the competence period, as exemplified by anchor cell ablation (*Félix, 2007*; *Milloz et al., 2008*), are more informative for model fitting and validation than

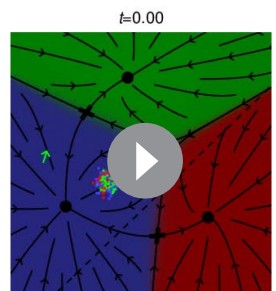

$t=0.00$

initially, Notch is weakly activated in all cells (arrow)

**Video 2.** Model dynamics for the JU2092 line with mild ectopic Notch and EGF ~3 xWT in *Figure 4C2*, showing how EGF can induce some P4/8.p to 2° fate while P5/7.p all remain 2° fate. At intermediate times, P5/7.p transiently pass through the 1° domain and produce sufficient Notch ligand to divert a fraction of P4/8.p from the 3° fate.

DOI: https://doi.org/10.7554/eLife.30743.018

end point data. However, other than these ablations, there is relatively little dynamic data to compare with. Timed temperature perturbations in temperature sensitive backgrounds clearly supply dynamic data and our model can predict the conditions under which dramatic effects are expected, *Figure 6*, *Videos 3–4*.

We showed in *Figure 5* that our model could explain the rescue of 1° fate induction by Notch in an EGF hypomorph. The predicted effect is very sensitive to timing, and can be realized by a temperature sensitive Notch mutant, *Figure 6A*. A Notch pulse early in induction diverts P6.p further away from the ridge between 3° and 1° than a late one, and thus induces more P6.p cells to the 1° fate.

The same holds for the converse effect, induction of 2° fates by weakly activating *lin-15 (ts)* in a background with weak ectopic Notch activity and no AC (*Zand et al., 2011*). The outcome is more dramatic if we take the same integrated level of EGF and concentrate it in the first quarter of the competence period, *Figure 6B*. This provides a direct test of our interpretation of pathway epistasis vs. Ras effector switching as described in *Zand et al. (2011)*.

Finally, we predict dramatic effects in the pure *lin-15(ts)* background, if we include P3.p in the model and then extend the duration of the heat pulse, *Figure 6—figure supplement 1*. Because the induced EGF is uniform across the VPCs, P3.p gets induced and provides an additional lateral signal to P4.p. Thus, there is a marked asymmetry between P4/8.p in the degree to which they are induced to 2° fates. To realize this state may require some tuning of the intensity of the temperature step, but the predicted 22123 pattern is noteworthy since it is induced by EGF and involves adjacent 2° fates.

Our fits imply that the EGF pathway is saturated in P6.p under WT levels. As a result, EGF activity and the rate at which P6.p progresses toward the 1° fate should not be too sensitive to moderate changes in EGF dosage. This prediction is consistent with the observed dynamics of Notch ligand expression in P6.p (an indicator of EGF activity and/or progression toward the 1° fate) under EGF overexpression, which is similar to WT (*van Zon et al., 2015*). The saturation of the EGF pathway can be tested by ablating the anchor cell in an EGF +/- background. We predict little change from the outcome for WT animals, in contrast with predictions for ablations in Notch perturbation lines (*Figure 6—figure supplement 2*).

## Fate correlations and multistability

Until this point, we have used a geometric model with no coupling between the two signaling pathways to explain epistasis in a variety of experiments. However, it is well known that Notch signaling inhibits the MAPK pathway and Ras activation inhibits Notch signaling (*Shaye and Greenwald, 2002*; *Berset et al., 2001*; *Yoo et al., 2004*). In what way could the predictions of the model be improved by adding terms to represent the known biochemical interactions within a cell?

A simple way to incorporate one such coupling in the model, with no additional parameters, is to down-regulate Notch signaling in 1° fated cells along the same threshold that defines the production of Notch ligands, cf. (*Shaye and Greenwald, 2002*). Such a term could schematically at least represent the down regulation of Notch receptors in 1° cells, and thus capture intra-cell pathway interactions that were not present in the base model that only described the (nonlinear) production of Notch ligands in 1° fated cells. Fitting this modified model to the same data, we find only minor adjustments to parameter values (*Table 2*), and the predictions are largely unchanged. The resulting phase diagram, *Figure 7B*, is very similar to *Figure 2A*. The most noticeable difference is in the

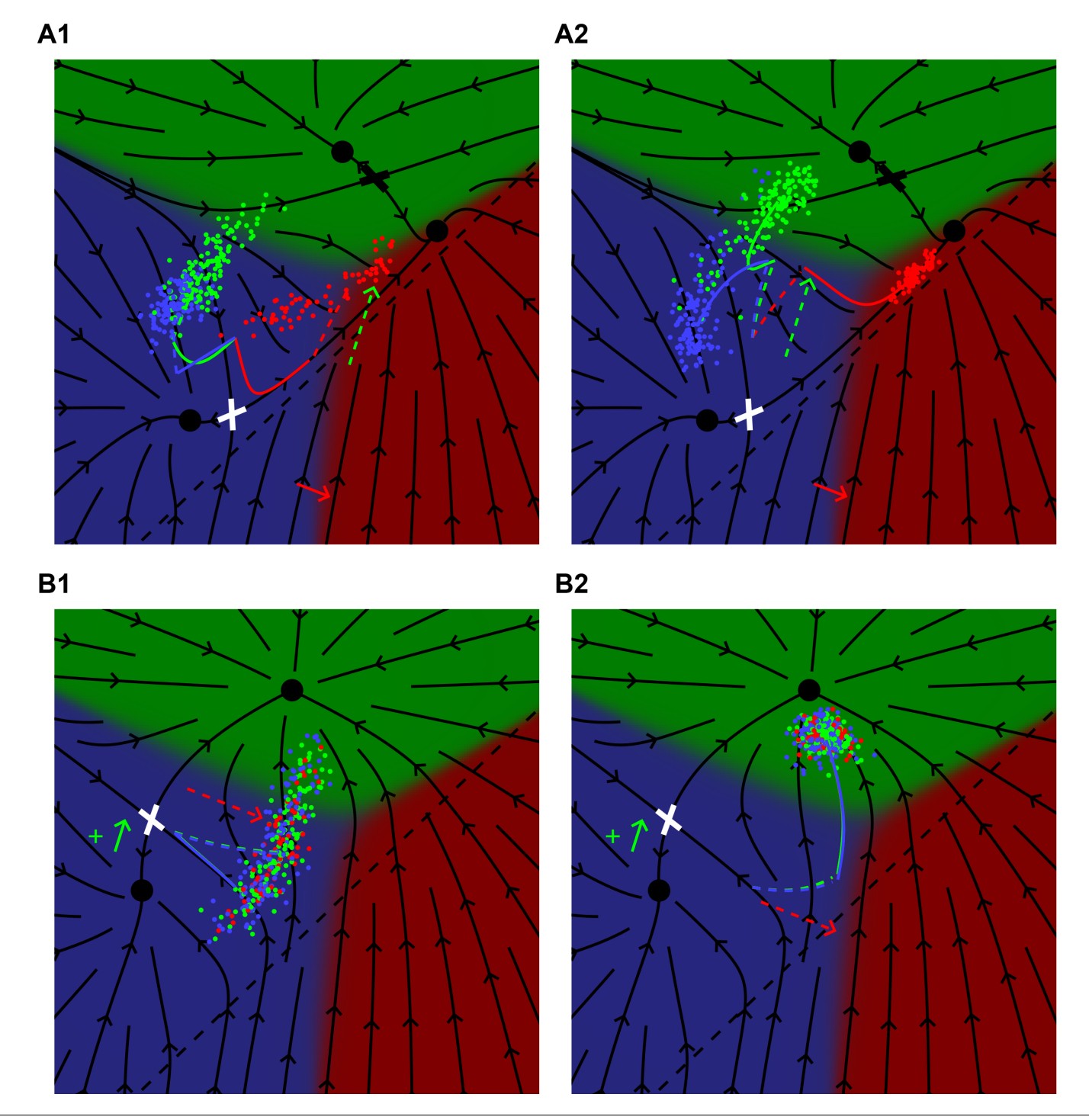

**Figure 6.** Model predictions for time-dependent perturbations. (**A**) A *lin-3(e1417)* hypomorph subject to a uniform ectopic pulse of Notch (WT + 0.4) late (0.75 < *t* < 1) (A1) or early (0 < *t* < 0.25) (A2) in the competence period. The black flow lines represent the behavior of P6.p in the absence of ectopic Notch signaling as defined in *Figure 5C1*. For a late Notch pulse, the average behavior of P6.p (red curve) follows the flow into the saddle point (white cross) and then out toward fate 1°. At *t* = 0.75 the red curve has a kink and is dashed to show how the Notch pulse moves the cells toward fates 2° and 3°. An early Notch pulse, A2, pushes the cells across the flow lines (dashed red), but at *t* = 0.25 there is a kink signifying the end of the Notch pulse, and cells then follow the flow and emerge in the 1° domain, showing nearly complete rescue of fate 1°. See *Video 3*. (**B**) A *lin-12* mutant with weak ectopic Notch activity (as defined in *Figure 5D1*) subject to a uniform pulse of EGF (WT + 0.64) either late (0.75 < *t* < 1) (B1) or early (0 < *t* < 0.25) (B2) in the competence period. The black lines show the dynamics in the absence of EGF following *Figure 5 D1*. In B1, the average cell

*Figure 6 continued on next page*

*Figure 6 continued*

(blue line) moves toward the saddle point as before, but at $t = 0.75$ has a kink, representing the onset of EGF signaling, and then (dashed) moves across the flow lines toward the 1° domain. The final outcome is a 73% to 25% ratio of 3° to 2° fates. With an early EGF pulse the trajectory (dashed) first moves toward fate 1° across the flow lines and at $t = 0.25$ has a kink representing the end of EGF signaling, and the trajectory (now solid blue) follows the flow and all cells assume fate 2°. See *Video 4*.

DOI: https://doi.org/10.7554/eLife.30743.020

The following figure supplements are available for figure 6:

**Figure supplement 1.** Transient lateral signaling yields adjacent 2° fates.

DOI: https://doi.org/10.7554/eLife.30743.021

**Figure supplement 2.** Anchor cell ablations in mutants.

DOI: https://doi.org/10.7554/eLife.30743.022

regime of reduced Notch and excess EGF: the 21112 domain extends to lower EGF levels, forming a longer boundary with the WT domain.

Why the two models differ in this regime is clear from the corresponding cell trajectories, *Figure 7C–F*. In the absence of coupling, moderate EGF overexpression ($1.5 \times$ WT) partially converts P5/7.p to the 1° fate, but the cells are clustered along the 1°−2° boundary and 1° fated cells do not generate sufficient Notch signal to divert P4/8.p from the 3° fate. Down-regulation of Notch in 1° fated cells restores a saddle point on the 1°−2° boundary that partitions the cells into distinct 1° and 2° groups, allowing the 1° cells to signal strongly to their neighbors. The resulting correlation between 1° fates in P5/7.p and 2° fates in P4/8.p is consistent with observations in the JU2113 line, with a slightly lower measured EGF level ($1.25 \times$ WT). The same bistability in P5/7.p implies a direct transition from a WT pattern to 21112, which is a strong prediction of our geometric modeling scheme, *Figure 7B*.

## Discussion

It is common in development to find a proliferating pool of progenitor cells that can be diverted into two mutually exclusive states by different signals. An example from early mammalian development is the split of the pluripotent epiblast cells into mesendoderm or neural fates at the time of gastrulation. Thus, our fate plane model with three possible fates is generic. The modeling proceeds from a general mathematical embodiment of the basic principles of development: cell fates are discrete, cells are competent to respond to signals over a limited period, and cells are committed when they assume their normal fates without additional signals. 'Fates' in the model are very schematic, they say nothing about the gene networks or morphogenetic events that ultimately define the fates. This simplification is plausible if the competence period ends before overt fate specification or differentiation occurs. The 'fates' in the model are then just a mathematical device to describe with minimal parameters all ways that two signals can divert cells towards three fates. Vulval patterning in *C. elegans* conforms to our assumptions (*Ambros, 1999*; *Wang and Sternberg, 1999*), and adds the simplification that the fates are terminal, so we can ignore subsequent bifurcations of the Waddington valleys.

Our fate plane represents the state of a cell. The motion of a cell in the plane will depend on signals impinging on that cell at the moment in question. Logically, the production of Notch ligand will depend only on where the cell is, not on the signals. Thus, we parameterized the onset of Notch ligand production by a line in the fate plane. Its extension far into the 3° fate domain is

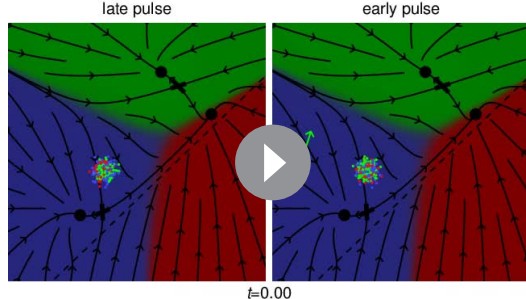

late pulse early pulse

$t=0.00$

ectopic Notch (green arrow) pushes P6.p across the flow lines

**Video 3.** Model dynamics for a *lin-3* hypomorph, subjected to a pulse of Notch late (left pane) or early (right pane) in the competence period following *Figure 6A* to show how timing affects outcomes. The flow lines represent the cell dynamics during the competence period in the EGF hypomorph. The green arrow shows when the Notch pulse is applied.

DOI: https://doi.org/10.7554/eLife.30743.023

late pulse    early pulse

$t = 0.00$

ectopic EGF (red arrow) pushes cells across the flow lines

**Video 4.** Model dynamics for weak ectopic Notch activity (with no AC and EGF signaling) subjected to an early (left pane) or late (right pane) spatially uniform pulse of EGF from *lin-15*(ts) showing how timing can affect outcomes, *Figure 6B*. The flow lines represent the cell dynamics during competence and in the absence of EGF. The EGF pulse (red arrow) pushes the cells toward the 1° domain.
DOI: https://doi.org/10.7554/eLife.30743.024

probably unreasonable, but since there is no situation that samples that region, it's immaterial.

The geometry that accompanies the Waddington landscapes informs the interpretation of genetic experiments. Since the model has three distinct states represented as valleys, there must be ridges between adjacent fates, with the property that with a little noise half the cells will go to each state. The aptly named saddle point is the low point on such a ridge. The topographic analogy makes it very evident that for cells positioned anywhere near the ridge that feeds into the saddle, a small initial perturbation can lead to large changes in outcomes that are generally impossible to predict without a model. A 'sensitized background' in genetics is precisely an allele that flows near a saddle point. Thus to infer the activity of a gene by crossing it into a sensitized background may require a model, and conversely such experiments are very informative for model fitting since they expose the boundaries between fates. *Figure 5* presented examples for vulva where small increments of Notch or EGF signaling in a sensitized background elicited non-intuitive changes in fates.

Our modeling approach yields the most parsimonious parameterization of how two signals can control three fates. The flow diagram is two dimensional to accommodate the two signals, there are three fixed points, one for each fate, and then mathematics requires the presence of two saddles. It is informative to scan the list of 12 parameters in *Table 2* that appear in the model. They all plausibly can be independently varied, for example, the time scale, noise level, EGF gradient, and autocrine signal all describe different phenomena. The other parameters define various vectors, also independent. The other models for vulva we discuss below have yet more parameters.

Several other models have been constructed for vulval patterning. Reference (*Fisher et al., 2007*) constructs a discrete model for the vulva system incorporating virtually all relevant genes. It was shown to be consistent by logical programming techniques. However, from our perspective it does not deal with partial penetrance or treat continuously variable morphogens. A differential equation built from selected genes with Michaelis-Menten dynamics was implemented in *Hoyos et al., (2011)* and required over 40 parameters. Because of the plethora of parameters both papers are reduced to making model inferences by assuming that the relative volumes of parameter space correctly weight embryo phenotypes. Closest to our approach is Reference (*Giurumescu et al., 2009*), who describe a cell with two coordinates (for Notch and EGF). They also do not fit partial penetrance data, or autocrine signaling, ignore the anchor cell ablation data that we found crucial for model selection, and also do not build in multistability. So formally at the end of competence all cells would revert to 3° fate. Nevertheless they use 13 dimensionful parameters, one more than we do.

We were able to describe the Notch and EGF signals as two vectors combined by vector addition since they appear inside of nonlinear functions that build in the basic landscape of the VPC dynamics, namely three fixed points and two saddles (Methods). This is why we assert that the inference of interactions is contingent on how we chose to represent the data, and the bulk of this paper is devoted to showing that what appeared as interactions when counting VPC fates, disappear when fit to our model as shown in *Figure 3–5*. An analogy with the physics of thermal equilibrium illustrates the point. If the data consists of concentrations, then to discover interaction energies between species one has to fit the logarithm of the concentrations, not the concentrations themselves. While there are no exact results in dynamics analogous to the Arrhenius formula, imposing the basic topology of the flow field on a linear model for the signals is a plausible place to begin.

Multiple steps in a signaling pathway are collapsed into one parameter; nevertheless, the model can capture intra-pathway interactions. In (*Corson and Siggia, 2012*), we described the cross between a loss-of-function mutant in the MAPK phosphatase, LIP-1 and ectopic EGF from *lin-15*, by

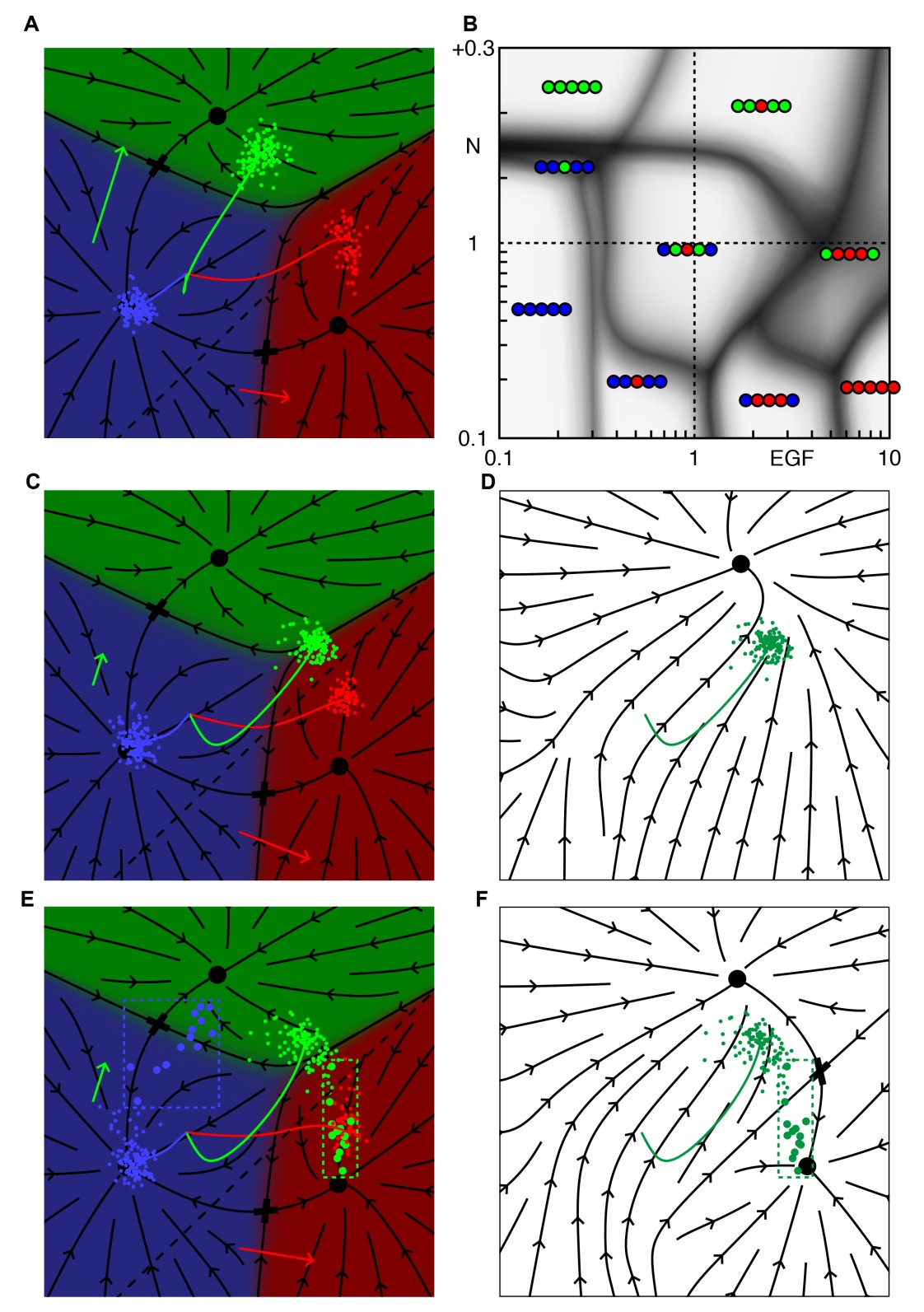

**Figure 7.** Inhibition of Notch in 1° fated cells clarifies the phase diagram and gives realistic correlated vulval patterns. (A-B) Notch signaling is down regulated along the same dashed line that defines production of Notch ligand (see Materials and methods). For WT parameters, A, 1° fated cells move further into the red domain (compare with *Figure 1E*). The modified phase diagram, B, has a well-defined 21112 domain for Notch below WT, and EGF ~5 (compare with *Figure 2A*). On the boundary with the WT domain, P5-7.p, all change fate simultaneously, demonstrating the correlation. C-F

*Figure 7 continued on next page*

*Figure 7 continued*

illustrate the relation between multistability and fate correlations along the boundary between WT and 21112 (Notch = 0.4 × WT and EGF = 1.5 × WT). The post competence flow diagram for our primary model is shown in C. The flow seen by P5/7.p near the end of the competence period (including autocrine Notch signaling) has a single attractor (D) and the outcomes are clustered near the 1/2° boundary (green dots in C,D, P4/8.p are not shown). If we allow for down regulation of Notch signaling in 1° fated cells, the saddle point (cross) between 1° and 2° is preserved (F), yielding outcomes for P5/7.p that are spread out (E). Then with some probability P5.p or P7.p signal strongly to their outer neighbors, inducing a 2° fate (the large dots in dashed boxes highlight animals where P5/7.p signal strongly to their neighbors).

DOI: https://doi.org/10.7554/eLife.30743.025

first adding the *lin-15* contribution to the EGF from the anchor cell and then multiplying the sum by a parameter larger than one to account for loss of the phosphatase. The order of composition is dictated by the genetics. Our model that takes a linear combination of the vectors parameterizing each pathway and passes the vector sum through a nonlinear function, fits the genetic data in (*Berset et al., 2001*) even though LIP-1 is a Notch target and biochemically inhibits EGF signaling. We do not question the biochemistry, but it is not needed to explain the genetic result. We do not have to introduce a new fitting parameter for the down regulation of the MAPK cascade by the LIP-1 phosphatase to explain the genetic data.

In the absence of fitting parameters explicitly coupling the two pathways, our model has reproduced effects previously attributed to pathway epistasis simply by including the basic genetic facts of vulval patterning. The most visible manifestations of intrinsic pathway interactions we found are correlations between the fates of adjacent cells. We were able to account for them schematically in *Figure 7*, without additional parameters, merely by assuming that the effects of Notch on 1° fated cells disappears when that cell produces Notch ligands. Mutual pathway inhibition builds determination into our model, that is, a cell that gets close enough to its terminal state cannot be deflected from that state by ectopic signals.

To go further, separate parameters would be needed for the down regulation of Notch by EGF and the inhibition of MAPK by Notch signaling (*Shaye and Greenwald, 2002*; *Berset et al., 2001*; *Yoo et al., 2004*), which we have not pursued since the necessary data are still very sparse, and this would have obscured effects that are independent of pathway interactions.

Intuitively, the most informative experiments to fit a dynamic model manipulate signaling during the competence period when the cell is poised among its fates. The timed anchor cell ablations were very instructive for the vulva model. With recent advances in microfluidics (*Keil et al., 2017*) one can follow larval development for up to three days and image the worms every 8 min. All the vulva divisions can be scored with DIC optics, and it is possible to control signals with temperature and ts-signaling alleles. In *Figure 6*, we presented instances where a Notch or EGF pulse delivered by a ts-mutant would have very different outcomes depending on the timing of the pulse within the competence period. These are appealing experiments since the intensity and duration of the pulse (provided it is much shorter than the competence window) will be adjusted to induce an appreciable phenotype, and then fit with one parameter. Then one quantifies the outcomes as a function of the phase of the pulse within the competence window with no additional parameters. The competence window itself can be redetermined by applying a temperature step to a ts-allele and the predictions are insensitive to details such as the shape of the pulse due to induction and persistence of the proteins (*van Zon et al., 2015*) that are impossible to control; only the relative timing matters.

The phase diagram is computed directly from the model, but its general structure can be deduced from qualitative features of the fate plane. Our vulva model in *Figure 1E* allowed direct contact between any pair of fates. If instead we assumed a fate plane with the 2° fate wedged between and separating the 1° and 3° fates (as might correspond to a graded action of EGF [*Katz et al., 1995*]), the WT pattern is preserved but the phase diagram is very different, *Figure 2— figure supplement 1*. There are no longer any triple points since three fates can never coexist in the fate plane. But ablation experiments (*Félix, 2007*; *Milloz et al., 2008*) required that the three valleys be adjacent in all combinations (*Corson and Siggia, 2012*), implying that phase boundaries meet at triple points. This and a few quantitative phenotypes largely determine the full diagram *Figure 2— figure supplements 2–4*. A phase diagram forces the recognition that individual phases have boundaries and that boundaries will in general meet at triple points. A model will assign coordinates

to all these features and allow experiments to be targeted to interesting points. Independent of a specific model, consideration of geometry unifies seemingly disparate outcomes.

Models that define the fate plane geometrically and then fit to genes can unify what appear to be different signaling modalities. The relative importance of induction of 2° vs. 'lateral inhibition' of 1° (*Sternberg, 1988*) by Notch has been debated. While biochemically the distinction may be very clear (e.g. when Notch signaling deactivates the MAPK pathway [*Berset et al., 2001*; *Yoo et al., 2004*]), once we insist fates 1° and 2° are distinct (bistability) there is no distinction between favoring one or inhibiting the other. While the extensive data in (*Barkoulas et al., 2013*) requires the authors to admit context-dependent signals (e.g. lateral inhibition vs. induction depends on EGF), our model accounts for all contexts with one set of parameters.

Similarly, vulval patterning is often conceptualized as an interplay of 'graded' (EGF level defines 1° vs. 2° fates [*Katz et al., 1995*; *Sternberg and Horvitz, 1986*]) and 'sequential' (Notch induces 2° fates [*Koga and Ohshima, 1995*; *Simske et al., 1995*]) signaling, which are proposed to be partially redundant (*Kenyon, 1995*). Our quantitative model suggests a specific instance of this: EGF reaching P5/7.p is not required for 2° fate induction but reduces loss of 2° fates under reduced Notch activity (*Corson and Siggia, 2012*).

Genes and their interactions are the ultimate building blocks of development, so how can a phenomenological model that eschews any one to one correspondence with genes contribute to our knowledge of development? For any signaling pathway, there are upwards of five components required from receptor to transcription and already 10's of parameters. There are solid grounds for believing most of these parameters do not matter for the behavior of the system (http://www.lassp.cornell.edu/sethna/Sloppy/). Signal transmission between cells is less understood, particularly for hydrophobic ligands such as Hedgehog. Though pathway components and their biochemical properties have been elucidated in cell culture systems, it is far from evident that actual parameters fit to those systems have any relevance to the embryo. If the goal is to be more quantitative about how an egg turns into an embryo, then it is not clear to us, how denser lists of genes and their interactions will help. Rather pieces of the problem will be parameterized and the components joined in a structure defined by phenomenology.

Evolution has built many levels of redundancy into development that insure a robust outcome but complicate efforts to reconstruct the process from its genetic components. Geometric models take a step back from genetic reductionism and impose basic embryological phenomenology (competence, commitment, determination etc.). They begin from a parsimonious representation of how signals define fates that largely eliminates redundancy among parameters, and leads to sharper fits and more believable predictions. Geometric models should be the method of choice when confronted with sparse in-vivo data in developmental biology (*Corson et al., 2017*).

# Materials and methods

## Model

The model (*Corson and Siggia, 2012*) describes the state of each VPC by a vector $\vec{r}$ in two-dimensional space and its dynamics by the stochastic differential equation

$$\frac{d\vec{r}}{dt} = \frac{1}{\tau}\left[\vec{\sigma}_1\left(\vec{f}+\vec{m}\right)-\vec{r}\right]+\vec{\eta}(t)$$

(1)

where

$$\vec{f}\left(\vec{r}\right) = 2\vec{r} - 2xy\vec{e}_x + \left(y^2-x^2\right)\vec{e}_y$$

(2)

is a polynomial vector field with threefold symmetry and the nonlinear function

$$\vec{\sigma}_1\left(\vec{f}\right) = \tanh\|\vec{f}\|\frac{\vec{f}}{\|\vec{f}\|}$$

(3)

ensures that the dynamics is bounded. The term

$$\vec{m} = \vec{m}_0 + l_1\vec{m}_1 + l_2\vec{m}_2 \tag{4}$$

integrates a default bias towards the default fate 3°, $\vec{m}_0$, and the effect of EGF and Notch signaling, parameterized by two vectors, $\vec{m}_1$ and $\vec{m}_2$ (red and green arrows in **Figure 1C**), which are combined linearly according to the levels of EGF and Notch ligands, $l_1$ and $l_2$, on the cell.

Variability in the dynamics is described by the stochastic term $\vec{\eta}(t)$, parameterized by a coefficient of diffusion $D$ in phase space:

$$\eta_i(t)\eta_j(t') = 2D\delta_{ij}\delta(t-t') \tag{5}$$

A fixed exponential gradient of EGF is assumed

$$l_1 = \{\gamma^2, \gamma, 1, \gamma, \gamma^2\} \tag{6}$$

while the level of Notch ligands a cell exposes to its neighbors is a function of its current state

$$L_2\left(\vec{r}\right) = \sigma_2\left(n_0 + \vec{n}_1 \cdot \vec{r}\right) \tag{7}$$

with

$$\sigma_2(u) = \frac{1 + \tanh(2u)}{2} \tag{8}$$

and varies continuously from 0 to 1 across a line parameterized by $n_0$ and $\vec{n}_1$ (dashed line in **Figure 1E**). The level of Notch ligands received by a cell integrates contributions from its neighbors and from autocrine signaling, e.g. for P6.p

$$l_2(\text{P6.p}) = L_2(\text{P5.p}) + \alpha L_2(\text{P6.p}) + L_2(\text{P7.}p) \tag{9}$$

where $\alpha$ is the relative strength of autocrine signaling. Modulations in signaling activity are represented as additive or multiplicative changes in the ligand levels. For example, EGF overexpression is described by

$$l_1 = \lambda\{\gamma^2, \gamma, 1, \gamma, \gamma^2\} \tag{10}$$

where $\lambda > 1$ denotes the dosage of EGF relative to WT, and ectopic EGF expression in a *lin-15* mutant (assumed uniform) by

$$l_1 = \{\gamma^2, \gamma, 1, \gamma, \gamma^2\} + \delta l \tag{11}$$

where $\delta l$ denotes the level of ectopic EGF.

Simulations proceed as follows. The VPCs are initially equivalent and their state is drawn from the (Gaussian) steady state distribution obtained in the model when the term $\vec{f}$ that makes the dynamics multistable is removed and there is no signaling (an ansatz for dynamics prior to VPC fate specification). The full dynamics are then computed for one unit of time, representing the period during which VPCs respond to EGF and Notch signaling, after which the fate of each cell is scored according to its final position. For this purpose, the dynamics is run for one more unit of time in the absence of signaling to allow convergence toward an attractor, then fractional fate assignments are computed according to the distance to the three attractors. With this fractional fate assignment, fate proportions output by the model under fixed realizations of the noise are continuous functions of model parameters, which is numerically convenient for parameter fitting.

It may be noted that the form of our equation for VPC dynamics bears a resemblance to standard models for gene expression, comprised of a sigmoidal production term and a linear degradation term. However, the equation is vectorial, and the two coordinates in phase space do not stand for the levels of particular molecular species, providing instead an effective representation of the progression of a cell toward its eventual fate.

## Parameter fitting

Model parameters are fit to a set of experimental data, that is, the proportions of the different fates, 1°−3°, adopted by each cell, P4-8.p, in different conditions (**Table 1**). In some backgrounds in which EGF signaling is perturbed, *lin-3* mRNA levels have been measured and are used as a proxy for EGF levels. Elsewhere, ligand levels are treated as unknown parameters and fit to the data.

Following the principles of Bayesian inference, we compute the posterior probability of a parameter set according to the deviation between fate proportions in the model (computed from multiple simulation runs) and in experiments, and to priors that disfavor values that are deemed biologically unreasonable, for example very large pathway sensitivities ($\vec{m}_1$, $\vec{m}_2$) or a very short response time ($\tau$) (**Corson and Siggia, 2012**).

Numerically, local maxima of the posterior probability are computed using the Levenberg-Marquardt algorithm. Several runs initialized from different, random parameter sets drawn from the prior distribution identify a unique global maximum. We then sample from the posterior using a Markov chain Monte Carlo algorithm initialized at the global maximum (**Corson and Siggia, 2012**).

The posterior probability of a parameter set $\Theta$ is given by

$$P(\Theta|D) \propto P(\Theta)P(D|\Theta) \tag{12}$$

where $P(\Theta)$ is the prior probability of $\Theta$ and $P(D|\Theta)$ is the likelihood of the experimental data $D$ given the outcome of the model with parameters $\Theta$.

In **Corson and Siggia (2012)**, we used the following approximation for the likelihood of the data,

$$P(D|\Theta) \approx e^{-\frac{\chi^2(\Theta)}{2}} \tag{13}$$

where

$$\chi^2(\Theta) = N \sum_{e,c,f} \left( p_{e,c,f}^{\text{exp}} - p_{e,c,f}^{\text{num}} \right)^2 \tag{14}$$

In this equation, $p_{e,c,f}^{\text{exp}}$ and $p_{e,c,f}^{\text{num}}$ denote the fraction of animals where cell $c$ adopts fate $f$ in experiment $e$, in experiments and in the model, respectively. $N$ is the typical number of animals and simulation runs per condition, set to 100. This inference procedure is intended to yield an approximate confidence interval for parameter values, and we use a uniform value of $N$ rather than actual animal counts, in order to give all phenotypes equal weight in the fit.

With the addition of new experiments to the data set (see below), we found that some experiments could be improperly fit. Specifically, for the JU1100 line where P4/8.p adopt the 1° fate in a small fraction of animals, the parameter ensemble obtained by Monte Carlo sampling included both parameter sets that match the phenotype, and parameter sets such that P4/8.p never adopt the 1° fate. Indeed, the 'cost' imposed by **Equation 14** for not fitting a small but non-zero $p_{e,c,f}^{\text{exp}}$ is small, which is an artifact of that approximate expression: the likelihood of the data, $P(D|\Theta)$, should be very small when this occurs. Here, we thus replaced **Equation 14** with

$$\chi^2(\Theta) = \frac{N^2}{4} \sum_{e,c,f} \frac{\left( p_{e,c,f}^{\text{exp}} - p_{e,c,f}^{\text{num}} \right)^2}{N p_{e,c,f}^{\text{num}} + \frac{1}{2}} \tag{15}$$

With the original data set from (**Corson and Siggia, 2012**), **Equation 15** yields a parameter distribution (means and standard deviations) that is very similar to that obtained using **Equation 14**. For the augmented data set used here, we obtain a better fit of the JU1100 phenotype.

## Augmented data set and comparison with original fit

Our model (**Corson and Siggia, 2012**) was fit to the vulval patterns observed in various mutants. But to be quantitative, it was essential that we used partially penetrant alleles and fit the observed variable outcomes. However, several of the relevant mutants did not have well-defined EGF levels. These could in principle be fit along with the phenotype, but as such only loosely constrained the model. Reference (**Barkoulas et al., 2013**) now has provided an abundance of quantitative data for defined EGF levels. While there are no qualitative discrepancies with our prior model, we can make

a much more stringent test of our modeling framework against the new data set if we add one of the newly quantified lines (JU1107) to the training data (*Table 1*) and then predict all the others. Doing this increases the parameter controlling the range of the EGF gradient ($\gamma$; cf. *Equation 6*) by about 50%. With the resulting flatter gradient, it takes a smaller increase in EGF dosage to obtain a phenotype. Our fit for the EGF level in the JU1100 overexpression line (*Hoyos et al., 2011*) (relative to WT) is now $4.2 \pm 0.5$ instead of $7.4 \pm 1.4$ This level was not measured and is a testable prediction of the model.

The second new piece of data we add to the training set is for epistasis between low EGF and Notch. We previously showed that model outcomes in this regime are sensitive to the directions of EGF and Notch signaling and can be synergistic (*Corson and Siggia, 2012*). The new data provides evidence for such a synergy (*Barkoulas et al., 2013*), and is accommodated by a slight adjustment of the directions, which differ by less than 20° from the default values we took in *Corson and Siggia (2012)* (*Table 2* gives all the parameter values and SD for the old and new fit).

## Incorporation of coupling between EGF and Notch

To incorporate downregulation of Notch signaling in 1° fated cells (*Shaye and Greenwald, 2002*) without introducing new parameters, Notch sensitivity is simply taken to vary inversely with a cell's production of Notch ligands. That is, we replace the term $\vec{m}$ describing the response to signals, cf. *Equations 1 and 3*, by

$$\vec{m} = \vec{m}_0 + l_1 \vec{m}_1 + \left( 1 - L_2 \left( \vec{r} \right) \right) l_2 \vec{m}_2 \tag{16}$$

The sensitivity of the cell to Notch signaling thus varies continuously from 1 to 0 as it crosses the dashed line in *Figure 1E*.

With this modification, the parameters of the model must be adjusted, and they are fit in the same way as for our main model.

## Acknowledgements

EDS was supported by NSF grant PHY-1502151. We thank W Keil, F Schweisguth, and S Shaham for comments. Some of this work was performed at the KITP Santa Barbara under National Science Foundation Grant No. NSF PHY-1125915

## Additional information

### Funding

| Funder | Grant reference number | Author |
| --- | --- | --- |
| National Science Foundation | PHY-1502151 | Eric D Siggia |
| National Science Foundation | PHY-1125915 | Francis Corson |

The funders had no role in study design, data collection and interpretation, or the decision to submit the work for publication.

### Author contributions

Francis Corson, Conceptualization, Software, Methodology, Writing—original draft, Writing—review and editing; Eric D Siggia, Conceptualization, Methodology, Writing—original draft, Writing—review and editing

### Author ORCIDs

Francis Corson  http://orcid.org/0000-0001-7230-137X
Eric D Siggia  http://orcid.org/0000-0001-7482-1854

### Decision letter and Author response

Decision letter https://doi.org/10.7554/eLife.30743.027

Author response https://doi.org/10.7554/eLife.30743.028

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
