## [Decision Letter]

Thank you for submitting your article "Gene free methodology for cell fate dynamics during development" for consideration by *eLife*. Your article has been favorably evaluated by Arup Chakraborty (Senior Editor) and three reviewers, one of whom is a member of our Board of Reviewing Editors. The reviewers have opted to remain anonymous.

The reviewers have discussed the reviews with one another and the Reviewing Editor has drafted this decision to help you prepare a revised submission.

The authors previously introduced a "geometric" dynamical model for *C. elegans* vulval development (Corson and Siggia 2012). A subsequent paper (Barkoulas et al. 2013) provided quantitative data for mutants perturbing the EGF and Notch pathways. In this manuscript, Corson and Siggia update their model, show that it accounts quantitatively for almost all observed phenotypes, and that it makes additional testable, non-obvious predictions.

The reviewers feel that this is strong and interesting work that merits publication in *eLife*, but we do have several issues and questions that we would ask you to address in a revised manuscript, listed in the following 9 points:

1) A main assertion is that this "geometric model with no coupling between the two signaling pathways (EGF and Notch) [is used] to explain epistasis in a variety of experiments". However, the model does couple EGF and Notch signaling pathways in several ways, including:

◦ While it is true that the two signals are parameterized by two vectors m→1and m→2, which are combined in a linear manner in Equation 4, the linear combination (vector m) is an argument to non-linear functions, including the hyperbolic tangent and the norm (Equation 3). This non-linear mapping will, in effect, introduce interaction terms involving products of m→1 and m→2. For example, consider the second-order term of the Taylor series expansion of the function tanh, or expand out the norm of l1m→1 + l2m→2, and one sees that there are products of components of vectors m→1 and m→2 in numerous places in the model equations.

◦ Making the above more complex to interpret, the interaction terms have no basis in mechanism.

◦ Next, the weighting factor (l2) for Notch signaling in Equation 4 depends on the level of EGF and Notch signaling, according to Equation 7 (note the dependence on the vector r, which depends on EGF and Notch signaling). Thus, the magnitude of Notch signaling depends on EGF signaling – a form of coupling.

◦ Finally, the threshold line for Notch expression (dotted line in Figure 1) cuts diagonally through the fate plane. This line represents yet another form of coupling, since the level of Notch ligand (l2) depends on passing this threshold line (which depends on EGF and Notch signaling).

2) It is asserted that this model framework is the "most parsimonious parameterization of how two signals can control three fates". This statement is debatable, and readily disproven by drawing examples from literature. Based on Table 2, there appears to be 12 parameters in the model. The footnote to the table indicates an additional one that was not fit (norm of vector 1→1), but is a parameter. In total, we have at least 13. There are many examples of developmental patterning models, including but not limited to this *C. elegans* system, with fewer parameters. Some of these other models even include molecular mechanism (coarse-grained), and are capable of predicting phase diagrams of multicellular phenotype. Therefore, parsimonious parameterization is probably not a major feature of the approach presented here. Rather, it seems more accurate to describe the model as taking a complex fate plane structure (Equations 1-11) and using numerous parameters to fit the dynamically-evolving landscape to experimentally-observed VPC fate choices. In this respect, the approach opens itself to a classical critique of phenomenological modeling in biology: given a sufficiently complex model structure, one can fit observed behavior to it.

3) There are a number of predictions about the dynamical behavior of the system (Figure 5–Figure 6, subsection “Dynamical perturbations are sensitive tests of the model”). However, it is unclear what confidence to have in these predictions because there is so little temporal data for this system with which to parameterize dynamics (as pointed out in the manuscript). For example, the dataset in [Barkoulas et al., 2012] is extensive, but appears to be only endpoint measurements. According to Table 1, it seems that the only data for intermediate time points came from anchor cell ablations in the wild-type worm ([S12]). While this data provides VPC fate choices (as shown in the table), there is no data on the level of Notch signal. Additional detail is needed on what temporal data is used to parameterize the model, how, and what degree of confidence one can have on the parameterization of dynamics.

4) This modeling approach provides a quantitative framework for predicting phenotypes, but is largely phenomenological. One may be left wondering what new things we learned about mechanisms involved in *C. elegans* vulval development. We are left with a black-box(*) of modulating EGF/Notch signal on one end, and post- or predicting phenotypes on the other end. Whether this is the most insightful avenue for modeling and drawing mechanistic insight into developmental systems is unclear. (*) perhaps a grey-box since the mathematical construction is precisely known to us. But, the construction is phenomenological and has no mechanistic basis.

5) Related #4, the notion that "geometric models should be the method of choice when confronted with sparse in-vivo data" seems an overstatement. Even in light of sparse in-vivo data and nascent mechanistic knowledge, significant progress has been made in many developmental model systems by using mechanistic models to predict phenotypes, and revising these models accordingly when experiments disprove model predictions.

6) There is little, if any, comparison to other models that predict phase diagrams for this system. Does this modeling approach match experimentally observed phenotypes better than other models? Do other models predict "sensitization" by quantitatively modulating specific signaling mechanisms? What are the pros and cons of the approach presented here relative to other methods?

7) Figure 1 show a dashed line in the fate plane, with an explanation in the legend that "cells positioned below the dashed line express Notch ligands." The dashed line appears invariable across the different panels. Does this mean that cells express Notch ligands regardless of their fate types (t=0.5 for P5/7.p for secondary fate, and several panels for tertiary fate)? If so, it will be good to cite experimental evidence for this.

8) In Figure 4C2, where authors predicts a transient primary fate for the green cloud (P5/7.p), the path is above the dashed line. How would these cells produce Notch ligands to transiently signal P4/8.p to induce secondary fate? I'm assuming that the paths are simulated results and not schematic drawings. If the latter, one may argue that the ectopic Notch is epistatic to the EGF, in which case one does not need to invoke a transition through the primary fate.

9) P6/P7/P8 are supposed to be an equivalence group, but Figure 1 shows P6.p with a different fate plane from P7.p/P8.p. Doesn't this beg the question? The model needs to explain how P6 becomes different from P7/P8. Figure 1 appears to be trying to illustrate the notion of competence windows (that P6.p has an early competence window to respond the EGF signal, and P7.p has a middle competence window to receive the Notch signal from P6.p). But it's unclear how the model treats competence windows. It seems the authors are trying to say that P6.p sees the highest/earliest EGF signaling from the AC, so P6.p starts "moving" in its fate plane first; and when P6.p starts expressing Notch, that starts moving P7.p. But it's not clear why the fate planes themselves change. If the EGF signal causes the fate plane of P6.p to change to an all-1* fate plane, doesn't this become tautological? (That is: the geometric model explains how P6.p adopts 1* fate by saying that P6.p moves toward 1* fate.)

[Editors' note: further revisions were requested prior to acceptance, as described below.]

Thank you for resubmitting your work entitled "Gene free methodology for cell fate dynamics during development" for further consideration at *eLife*. We apologize for the delay in examining your revised manuscript. Your revised article has been favorably evaluated by Arup Chakraborty (Senior Editor), and a Reviewing Editor.

The manuscript has been greatly improved, and the referees agree that most concerns have been satisfactorily addressed. However, there is one aspect (1a, b below) that needs to be addressed before a decision to accept the paper can be made, because a central claim of the manuscript remains unclear. There is also one other aspect (2 below) that the referees note for you to consider at your discretion.

1a) Regarding whether the model includes coupling between EGF and Notch, your response acknowledges "couplings induced between the N and EGF pathways by the non-linear functions that are used". However, the manuscript still states in several places that E and N are not coupled in the model, including:

"Our model has no overt coupling between the EGF and Notch pathways, implying that if we fit two alleles,[…]".

"Until this point, we have used a geometric model with no coupling between the two signaling pathways[…]"

"Our model with no pathway interactions then fits the genetic data in…"

You are trying to make a distinction between "overt" versus implicit coupling. Whether the coupling is overt or implicit, however, the end result is that the signals are coupled with each other, and the manuscript should not claim that they are not.

1b) In addition to coupling introduced by nonlinear functions, the model includes coupling in its geometric construction. This point was in the original review (see final bullet under point 1) but left unaddressed. Consider Figure 7 and subsection “Fate correlations and multistability”, where it says "a simple way to incorporate one such coupling in the model, with no additional parameters, is to down-regulate Notch signaling in 1º fated cells along the same threshold [the dotted line in Figure 1] that defines the production of Notch ligands". If downregulating Notch signaling along a line in the fate plane is a way to introduce EGF-Notch coupling, would not upregulating/increasing the production of Notch ligands as cells cross the same line also be a form of coupling? The latter is part of the model from the start.

This coupling has nothing to do with the shape/steepness/non-linearity of the threshold – it is a coupling introduced by a demarcation in the fate plane at which Notch ligands are unregulated. To cross that demarcation, cells must possess a combination of EGF and Notch signaling that lands them in that place in the phase plane. Thus, the level of EGF and Notch signaling affects expression of Notch ligand, which in turn affects EGF and Notch signaling: geometric coupling.

This is a second reason to question the manuscript's central claim to predict epistasis without invoking coupling; please clarify.

2) The comparisons to other models were useful. Regarding model complexity as quantified by numbers of parameters – the manuscript still speaks of a vague "plethora of parameters", but you provided useful parameter numbers for the other models in your response (over 40 parameters (Hoyos) and 13 parameters (Giurumescu)). It would clarify the scale of the disparity if you included these numbers in the manuscript.

---

## [Author Response]

[…] The reviewers feel that this is strong and interesting work that merits publication in eLife, but we do have several issues and questions that we would ask you to address in a revised manuscript, listed in the following 9 points:1) A main assertion is that this "geometric model with no coupling between the two signaling pathways (EGF and Notch) [is used] to explain epistasis in a variety of experiments". However, the model does couple EGF and Notch signaling pathways in several ways, including:◦ While it is true that the two signals are parameterized by two vectors m→1 and m→2, which are combined in a linear manner in Equation 4, the linear combination (vector m) is an argument to non-linear functions, including the hyperbolic tangent and the norm (Equation 3). This non-linear mapping will, in effect, introduce interaction terms involving products of m→1 and m→2. For example, consider the second-order term of the Taylor series expansion of the function tanh, or expand out the norm of l1m→1 + l2m→2, and one sees that there are products of components of vectors m→1 and m→2 in numerous places in the model equations.◦ Making the above more complex to interpret, the interaction terms have no basis in mechanism.◦ Next, the weighting factor (l2) for Notch signaling in Equation 4 depends on the level of EGF and Notch signaling, according to Equation 7 (note the dependence on the vector r, which depends on EGF and Notch signaling). Thus, the magnitude of Notch signaling depends on EGF signaling – a form of coupling.◦ Finally, the threshold line for Notch expression (dotted line in Figure 1) cuts diagonally through the fate plane. This line represents yet another form of coupling, since the level of Notch ligand (l2) depends on passing this threshold line (which depends on EGF and Notch signaling).

We fully agree with the bulleted remarks of the referees pointing to couplings induced between the N and EGF pathways by the non-linear functions that are used to saturate the response and create the correct topology (i.e., saddles and fixed points) of the model. This is a feature not a bug, as one says in the software field!

Our intention is best expressed by an analogy. If one is given data for the concentrations of various components in a system in thermal equilibrium, then it is most sensible to fit the *log* of the concentrations to free parameters *not* the concentrations themselves. This is because the Arrhenius formula shows that the log is necessary to extract *interaction energies* that are the most compact representation of statistical mechanics. Thus we remark in the Discussion regarding Berset et al., 2001, that their genetic data does not require pathway interaction (according to our model, but contrary to their assertions), though they supplied biochemical evidence that N inhibits EGF signaling, which of course we cannot question. A most important consequence of our paper, and why we hope it reaches a wide biological audience, is that many assertions about genetic interactions in the literature are not well founded, precisely because of the choice of variables used to make the comparison.

Although it’s very unlikely that some exact decomposition of non-equilibrium systems exists, one’s first guess as to what the analogue of Arrhenius would be is to build in the correct topology of the developmental system in question and then ‘tilt’ the landscape with linear functions. This is certainly a well-defined procedure and we show it suffices to fit the available data. It is universal in neural nets for machine learning, to subject a linear combination of inputs to a nonlinear thresholding function with one output. However in biological problems we claim the nonlinear function should encapsulate the topology of system. In our 2012 PNAS paper we showed that a topology of three fixed points and two saddles in one dimension was qualitatively wrong for the vulva and our model is the next more complex one. Topologies are discrete and just as one does in fitting data with a polynomial, the simplest one that works wins.

Evidently we have not been clear about what to us is the general message of our paper. We thus added a paragraph to the Discussion where we pose the question of induced interactions via the nonlinear functions (raised by the referees), make the Arrhenius analogy, and give some intuition why our procedure for a linear representation of the signaling pathways subject to a nonlinear thresholding function dictated by the topology is the optimal way to model developmental transitions.

2) It is asserted that this model framework is the "most parsimonious parameterization of how two signals can control three fates". This statement is debatable, and readily disproven by drawing examples from literature. Based on Table 2, there appears to be 12 parameters in the model. The footnote to the table indicates an additional one that was not fit (norm of vector n→1), but is a parameter. In total, we have at least 13. There are many examples of developmental patterning models, including but not limited to this C. elegans system, with fewer parameters. Some of these other models even include molecular mechanism (coarse-grained), and are capable of predicting phase diagrams of multicellular phenotype. Therefore, parsimonious parameterization is probably not a major feature of the approach presented here. Rather, it seems more accurate to describe the model as taking a complex fate plane structure (Equations 1-11) and using numerous parameters to fit the dynamically-evolving landscape to experimentally-observed VPC fate choices. In this respect, the approach opens itself to a classical critique of phenomenological modeling in biology: given a sufficiently complex model structure, one can fit observed behavior to it.

We intended our statement to be provocative but it is close to the truth.

The first four parameters in Table 2 are the two dimensional vectors representing the N and EGF influence on the state of a cell. Representing a signaling pathway (from receptor to fate and thus implicitly transcriptional output) with two parameters, is certainly parsimonious and given our two dimensional phase plane, nothing less is possible. A gene level description would have Michaelis-Menten expression for ligands to receptors, receptors phosphorylate effectors, and perhaps the MAPK pathway for EGF etc.

The fifth parameter is a time scale, that determines the time needed for induction relative to the duration of the competence period, as probed by AC ablation experiments.

The diffusion constant largely controls the extent of the partial penetrance of many of the phenotypes we fit, hence manifestly necessary.

Next is how the EGF gradient from the Anchor Cell falls off in space. Clearly independent of other parameters.

Autocrine N signaling, is the one parameter that allows P6.p to assume some 2 fate under early AC ablation.

The remaining two parameters (or three if the referees insist) define the domain in the fate plane where the cell turns on N ligand to signal to itself and its neighbors. A line is the simplest way of dividing a plane. Notch turns on smoothly (via a tanh function) as cells approach the dashed line in Figure 1. The width of the onset is not constrained by the data and not fit. This is because cells will continue to approach this line, till they are pushed upwards by the Notch signal; changing the width of the transition will alter slightly when they reflect off the line, but not their ultimate fate. Hence 11 parameters in Table 2 in all, (Note none of the data constrains the angle of m_0_, points in that region always relax towards the fixed point, so we just left it at a default value). Thus we claim each parameter represents a non-redundant piece of any model that will describe comparable phenomena. We give error bars on the parameters in Table 2 from Monte Carlo simulations and all are modest, hence all parameters are constrained by the data.

The referees correctly ask us in #6 to mention other models of vulval patterning, so we can count parameters in these papers:

Hoyos..Felix 2011. An ODE model with 11 species, and over 40 parameters, as one would expect for Michaelis-Menten dynamics.

Closer to our work is Giurumescu..Sternberg 2009 in that they represent each cell with two variables for EGF and N taken to define the coordinate axes. They have 13 dimensional parameters, 9 dimensionless (see below their Equation 6) but they do not treat partial penetrance, hence omit our D, and do not consider experiments revealing autocrine signaling. Hence we are more parsimonious 11 – 2 < 9 + 1 (counting one absolute time scale for each model). More important their model is not multi-stable hence there is no specification/commitment, remove the external signal and the cells collapse to fate 3.

Fisher..Hajnal 2007 build a discrete model with virtually all genes known. They have’ 92,000 possible reachable states’, but no real numbers.

Evidently we succeeded in being provocative without being cogent. So we have qualified this statement that our parsimony is relative to other models (real and imaginable) of vulval patterning, and then explain briefly why the various fitting parameters are non-redundant. Clearly a model with relaxation in a 1D potential with 3 minima would have fewer parameters, but would not encode the salient facts of the vulva system (a point made in our 2012 PNAS paper and mentioned again here).

3) There are a number of predictions about the dynamical behavior of the system (Figure 5–Figure 6, subsection “Dynamical perturbations are sensitive tests of the model”). However, it is unclear what confidence to have in these predictions because there is so little temporal data for this system with which to parameterize dynamics (as pointed out in the manuscript). For example, the dataset in [Barkoulas et al., 2012] is extensive, but appears to be only endpoint measurements. According to Table 1, it seems that the only data for intermediate time points came from anchor cell ablations in the wild-type worm ([S12]). While this data provides VPC fate choices (as shown in the table), there is no data on the level of Notch signal. Additional detail is needed on what temporal data is used to parameterize the model, how, and what degree of confidence one can have on the parameterization of dynamics.

We agree fully with the referee’s remarks that the only dynamical data are the anchor cell ablations from Felix that were exceedingly tedious to obtain but very valuable (see our 2012 PNAS paper). There are also no good fluorescent reporters for cell fate, e.g., egl-17 does not correlate well with primary fate in mutant backgrounds. Endpoint data is typically very insensitive to the nonlinear dynamics that led there, a point to be made forcefully about genetic screens that assay the mutated embryos well after the genes under study were active. However our model, after fixing an overall time scale, makes predictions for dynamics with no free parameters, hence this becomes a strong test of the model.

Our strategy to address these problems begins with a new microfluidic device built by a postdoc working with Siggia and Shaham (Keil, Dev Cell 2016), that permits multiday larval imaging. (This device is unprecedented in the field as evidenced by 3 collaborations Keil has entered with prominent worm labs outside of Rockefeller.) We are using ts alleles to deliver pulses of N or EGF to a population of worms in parallel. We have complete temporal control of the temperature. The worms are not precisely synchronized in the device, but we can define their developmental phase relative to competence by recording the divisions (which we can do with 8 min accuracy down to the last one).

We emphasized potential experiments in Figure 6 where there were dramatic changes in outcome depending where within the competence periods a stereotyped pulse of N or EGF was applied. Our apparatus will manifestly achieve these conditions. We will tune the temperature to see an effect, and the N or EGF level realized becomes a fitting parameter in the model. But the prediction is how the outcome changes with the phase of the pulse relative to the competence window. (The competence window itself can be redetermined with temperature steps followed by a record of when and how the cell subsequently divides.) The shape of the pulse does not matter and is folded into the fit of the amplitude. One pulse is applied to worms randomly scattered over the competence window. There are no free parameters once the pulse amplitude is fit.

We have substantially increased the paragraph in the Discussion that deals with future dynamic experiments to address these questions.

4) This modeling approach provides a quantitative framework for predicting phenotypes, but is largely phenomenological. One may be left wondering what new things we learned about mechanisms involved in C. elegans vulval development. We are left with a black-box(*) of modulating EGF/Notch signal on one end, and post- or predicting phenotypes on the other end. Whether this is the most insightful avenue for modeling and drawing mechanistic insight into developmental systems is unclear. (*) perhaps a grey-box since the mathematical construction is precisely known to us. But, the construction is phenomenological and has no mechanistic basis.5) Related #4, the notion that "geometric models should be the method of choice when confronted with sparse in-vivo data" seems an overstatement. Even in light of sparse in-vivo data and nascent mechanistic knowledge, significant progress has been made in many developmental model systems by using mechanistic models to predict phenotypes, and revising these models accordingly when experiments disprove model predictions.

We believe the community has been overly sanguine about what has/can be achieved by ‘mechanistic models’ in a developmental context, where we interpret ‘mechanistic’ to mean models where the variables correspond to the proteins, messages, regulatory DNA etc. in question. Although our remark has most force in development i.e., an embryo, it’s still relevant to single cell systems.

The basis for our pessimism about mechanistic models is illustrated by the Wnt pathway homepage that R. Nusse maintains. For just Wnt reception in a cell, there are about 8 components in a minimal model and perhaps another 40 have known interactions with the core 8, and this merely for signal reception. Any ‘mechanistic’ model represents composite variables.

The problem is actually more severe since a colleague Jim Sethna (http://www.lassp.cornell.edu/sethna/Sloppy/) has shown in great detail that models fit to pathway data for a single cells really only have a few well defined parameters, the others are ‘sloppy’, completely unconstrained by the data. This is shown mathematically from the eigenvalues of the Hessian of the error function being minimized.

We are interested in embryos. We do not accept that rates measured in a random cell line have any relevance to an embryo, hence there is exceedingly little quantitative embryo data, and even for the extraordinarily well-studied vulva we have to resort to phenomenological geometric models. If some mechanism is retained, it should be in the context of a phenomenological model to cover the rest of the system.

We have now incorporated the above discussion, and particularly the sloppy model connection, into a new penultimate paragraph exposing more fully issues living below the surface of mechanistic models. Our final sentence about “geometric models method of choice” will be restricted to developmental problems.

The referees under #4 also raise an interesting epistemological question, of whether a phenomenological model, can possibly add to knowledge since the atoms of knowledge in development (we infer from those remarks) must be phrased in terms of genes and their interactions. We would argue (following the Hegelian synthesis-antithesis path), that the very success of developmental genetics for the reasons above requires we reconceptualize developmental biology. A vast spread-sheet of all genes and their interactions is information rather than knowledge. More modestly if ‘knowledge’ implies some degree of quantitative prediction, that with stem cells will allow one to build embryos and organs, then we maintain that phenomenology is a prerequisite to understanding.

The penultimate paragraph of the Discussion rephrases the above argument. If any of the referees or editors find it too preachy, pretentious, or opinionated, we are happy to excise the paragraph in the editorial or proofing stage of the manuscript. It may be a good idea to do so.

6) There is little, if any, comparison to other models that predict phase diagrams for this system. Does this modeling approach match experimentally observed phenotypes better than other models? Do other models predict "sensitization" by quantitatively modulating specific signaling mechanisms? What are the pros and cons of the approach presented here relative to other methods?

This was an omission, we have added a discussion of the 3 papers mentioned under item 2, since they cover the different prototypes of modeling strategies. Prior theories never had the means to fit quantitative partial penetrance data, which is for us is the most revealing since it defines the boundaries between fates. There is no discussion of sensitized backgrounds as flows near saddle points, since partial penetrance is not allowed from the start. Further details are in the new Discussion.

7) Figure 1 show a dashed line in the fate plane, with an explanation in the legend that "cells positioned below the dashed line express Notch ligands." The dashed line appears invariable across the different panels. Does this mean that cells express Notch ligands regardless of their fate types (t=0.5 for P5/7.p for secondary fate, and several panels for tertiary fate)? If so, it will be good to cite experimental evidence for this.

This is a fair criticism, since there is biochemical data showing that the signaling pathways interact at the level of signal reception e.g., N signaling induces a MAKP phosphatase or EGF inhibits N signaling. However the dashed line defines where in the fate plane of the cell N ligand is produced. Logically for us, the state of the cell i.e., the point in the fate plane, defines whether that cell produces Notch ligands. Our assumption that the fate plane is 2D has a lot of content. We do not know if a cell in the 3º fated domain were to suddenly receive a huge ectopic EGF signal, would secrete Notch ligands before it moved to the relevant region of the phase plane. The model would say no, but there is no data. (Note the large green domain at T=0.5 for P5/7.p mentioned by the referee means only that under those signaling conditions, all cells move towards fate 2, not that they are already fate 2). So as elsewhere, the dashed line is the minimal fitting parameter, with the same logical status as the two vectors representing N and EGF signaling. Where it extends deep into the 3º domain it’s probably incorrect, but there is no data in that region, and no way we can see for forcing cells there.

We have not added these clarifications to the first paragraphs of Results where we introduce the model, to keep the Introduction as simple as possible. They are picked up in Discussion when talking about the minimality of the model.

8) In Figure 4C2, where authors predicts a transient primary fate for the green cloud (P5/7.p), the path is above the dashed line. How would these cells produce Notch ligands to transiently signal P4/8.p to induce secondary fate? I'm assuming that the paths are simulated results and not schematic drawings. If the latter, one may argue that the ectopic Notch is epistatic to the EGF, in which case one does not need to invoke a transition through the primary fate.

As we mentioned under #2 above, the dashed line is the center of a smooth ramp whose width (the additional parameter |n_1_| in question #2) that we did not fit. The points shown are actual simulations, and during their trajectory (more so than at the endpoint) P4/8.p have moved far enough onto the ramp representing N production that the resulting signaling together with ectopic N in that background directs the P4/8.p cells upwards into the fate 2 domain. There would be a minimal change in outcomes if we made reasonable changes in the width of that domain. The ectopic Notch in this strain, would not in isolation lead to the conversion of P4/8.p to fate 2, it required the added EGF to push P5/7.p close to the dashed line and generate additional N signal.

9) P6/P7/P8 are supposed to be an equivalence group, but Figure 1 shows P6.p with a different fate plane from P7.p/P8.p. Doesn't this beg the question? The model needs to explain how P6 becomes different from P7/P8. Figure 1 appears to be trying to illustrate the notion of competence windows (that P6.p has an early competence window to respond the EGF signal, and P7.p has a middle competence window to receive the Notch signal from P6.p). But it's unclear how the model treats competence windows. It seems the authors are trying to say that P6.p sees the highest/earliest EGF signaling from the AC, so P6.p starts "moving" in its fate plane first; and when P6.p starts expressing Notch, that starts moving P7.p. But it's not clear why the fate planes themselves change. If the EGF signal causes the fate plane of P6.p to change to an all-1* fate plane, doesn't this become tautological? (That is: the geometric model explains how P6.p adopts 1* fate by saying that P6.p moves toward 1* fate.)

This is a problem with our figure, and more generally using a static figure to stand in for a movie. Hopefully with the web centric *eLife* presentation it all will be clear. In Figure 1 we should have labeled the first row as t=0+, meaning just after the EGF signal turned on. Prior to that time the movie makes very clear that P4-8.p are completely equivalent (modulo the molecular noise) and bouncing around in the same potential. We model the competence window with hard boundaries, i.e. EGF reception jumps from 0 to 1 at t=0 and then off at t=1, since there is no data that is sensitive to what must be a continuous onset and decay of signaling. The last row of Figure 1 should be labeled t=1+, i.e., the signal turns off simultaneously for all cells and the flow fields are identical for P4-8.p though where the cells are is different and reflects their histories. Thus as expected P4-8.p are completely equivalent prior to competence and the same competence window applies to all cells.

We have modified the time labels on Figure 1 to add the ‘+’. Our meaning was defined in the caption, which in the review pdf was separated from the figure. Hopefully on line and with the linked video, or on a properly formatted pdf the equivalence group and common competence window all will be clear.

[Editors' note: further revisions were requested prior to acceptance, as described below.]

The manuscript has been greatly improved, and the referees agree that most concerns have been satisfactorily addressed. However, there is one aspect (1a, b below) that needs to be addressed before a decision to accept the paper can be made, because a central claim of the manuscript remains unclear. There is also one other aspect (2 below) that the referees note for you to consider at your discretion.

Our terminology is borrowed from conventional usage in *C.elegans* (e.g., http://wormbook.org/chapters/www_vulvaldev/vulvaldev.html, Section 8 coupling of LET-23 and LIN-12) where ‘coupling’ means intracellular interaction between EGF and N. For instance activation of the N receptor in a cell, induces a phosphatase that deactivates the MAPK pathway stimulated by the EGF receptor in the same cell. We adhere to an analogous use of the term in describing how our model fits data.

There are no new fitting parameters devoted to pathway interactions (until Figure 7 where there is interaction with no new parameters). Operationally this means that if we fit allele 1 for EGF and allele 2 for N, then we have no additional freedom to fit an animal with both alleles. Hence our paper was devoted to the challenge of fitting a matrix of conditions in the Barkoulas paper with parameters that depended individually only on the two axes. More colloquially N^2 phenotypes are fit with 2N parameters.

We thus consider it an achievement to demonstrate that a model with parameters only for the two pathways separately and independently, fits data that by phenotype shows interaction between the pathways. We need make this distinction clear to the readership of *eLife* and not just the worm community. Thus we propose the following changes in response to the referee report:

1a) Regarding whether the model includes coupling between EGF and Notch, your response acknowledges "couplings induced between the N and EGF pathways by the non-linear functions that are used". However, the manuscript still states in several places that E and N are not coupled in the model, including:"Our model has no overt coupling between the EGF and Notch pathways, implying that if we fit two alleles,…".

New text:Our model has no fitting parameters that would couple the EGF and Notch pathways, implying that if we fit two alleles, one in each pathway, then the outcome of the cross is defined with no additional freedom. […] The model produces a context dependent association between signals and fates since it applies nonlinear functions to a linear combination of the two vectors representing the pathways.

"Until this point, we have used a geometric model with no coupling between the two signaling pathways[…]"

New text:Until this point, we have used a geometric model with no fitting parameters that couple the two signaling pathways to explain epistasis in a variety of experiments. However it is well known that Notch signaling inhibits the MAPK pathway and Ras activation inhibits Notch signaling [...]. In what way could the predictions of the model be improved by adding terms to represent the known biochemical interactions within a cell?

"Our model with no pathway interactions then fits the genetic data in[…]"

New text:Our model, that takes a linear combination of the vectors parameterizing each pathway and passes the vector sum through a nonlinear function, fits the genetic data in [Berset et al., 2001] even though LIP-1 is a Notch target and biochemically inhibits EGF signaling. We do not question the biochemistry, but it is not needed to explain the genetic result. We do not have to introduce a new fitting parameter for the down regulation of the MAPK cascade by the LIP-1 phosphatase to explain the genetic data.

You are trying to make a distinction between "overt" versus implicit coupling. Whether the coupling is overt or implicit, however, the end result is that the signals are coupled with each other, and the manuscript should not claim that they are not.1b) In addition to coupling introduced by nonlinear functions, the model includes coupling in its geometric construction. This point was in the original review (see final bullet under point 1) but left unaddressed. Consider Figure 7 and subsection “Fate correlations and multistability”, where it says "a simple way to incorporate one such coupling in the model, with no additional parameters, is to down-regulate Notch signaling in 1º fated cells along the same threshold [the dotted line in Figure 1] that defines the production of Notch ligands". If downregulating Notch signaling along a line in the fate plane is a way to introduce EGF-Notch coupling, would not upregulating/increasing the production of Notch ligands as cells cross the same line also be a form of coupling? The latter is part of the model from the start.

The referee has thoroughly understood the content of the model. In the material quoted above, we were making a distinction between the production of Notch ligands in 1 fated cells, so something secreted and contained in the base model, versus a new term that down regulates Notch signaling within a 1 fated cell. We considered the new term, ‘interaction’ because it would represent biochemistry happening within a cell. We claim there is a logical distinction here, but evidently to a careful reader the term ‘interaction’ does not convey our meaning. We propose the following changes to two paragraphs in the new draft.

New text:[…] “In what way could the predictions of the model be improved by adding terms to represent the known biochemical interactions within a cell?[…] Such a term could schematically at least represent the down regulation of Notch receptors in 1º cells, and thus capture intra-cell pathway interactions that were not present in the base model that only described the (nonlinear) production of Notch ligands in 1º fated cells.”

This coupling has nothing to do with the shape/steepness/non-linearity of the threshold – it is a coupling introduced by a demarcation in the fate plane at which Notch ligands are unregulated. To cross that demarcation, cells must possess a combination of EGF and Notch signaling that lands them in that place in the phase plane. Thus, the level of EGF and Notch signaling affects expression of Notch ligand, which in turn affects EGF and Notch signaling: geometric coupling.

We agree with the referee’s interpretation of the model, and thus have revised our language in various places. (e.g. see the second paragraph of the subsection “Fate correlations and multistability”).

This is a second reason to question the manuscript's central claim to predict epistasis without invoking coupling; please clarify.

We concede that our prior terminology of ‘no pathway interactions’ and ‘predict epistasis’ could be read as an oxymoron. We hopefully have clarified that by interaction we mean ‘new fit parameters describing intra-cell events that are specific to a particular combination of the two pathways’.

We have searched our draft for all occurrences of ‘coupling’ and ‘interaction’ and verified that all associated remarks clearly convey the above meaning, and nothing more. The authors are happy to clean up their use of language.

2) The comparisons to other models were useful. Regarding model complexity as quantified by numbers of parameters – the manuscript still speaks of a vague "plethora of parameters", but you provided useful parameter numbers for the other models in your response (over 40 parameters (Hoyos) and 13 parameters (Giurumescu)). It would clarify the scale of the disparity if you included these numbers in the manuscript.

We wanted to be less confrontational regarding prior work in our text so were vague about number of parameters. We have rephrased those sentences to state the number of parameters employed in prior work.